# Understanding and Improving Length Generalization in Hierarchical Sparse Attention Models

**Jiaqi Leng**[1*]  **Xiang Hu**[2* †]  **Junxiong Wang**[3]  **Jianguo Li**[4]  **Wei Wu**[5†‡]  **Yucheng Lu**[6‡]

[1]Fudan University  [2]Tencent AI Lab  [3]Cornell University
[4]Ant Group  [5]Ant International  [6]NYU Shanghai

jqleng22@m.fudan.edu.cn, shawnxxxhu@tencent.com, lu.yucheng@nyu.edu

 github.com/jacky-leng/length-generalizable-sparse-attention

## Abstract

Effectively processing long contexts is a critical challenge for language models. While standard Transformers are limited by quadratic complexity and poor length extrapolation, alternative architectures like sliding window attention and state space models sacrifice the ability to effectively utilize the full context due to their fixed-size memory. Chunk-based sparse attention has emerged as a promising paradigm for extreme length generalization, yet the key architectural principles underpinning its success are not yet fully understood. In this work, we present a systematic dissection of these models to identify the core components driving their performance. Through a unified framework and comprehensive ablation studies, we demonstrate that a combination of three design principles is critical: (1) an expressive, non-linear Chunk Encoder with a dedicated CLS token to produce representations for retrieval; (2) a Bypassing Residual Path to stably integrate retrieved global information without it being overridden by the local residual stream; and (3) enforced selection sparsity during pre-training to bridge the train-test distribution gap. We provide a theoretical motivation for intra-chunk information processing and landmark generation. By combining these principles, we establish a new state-of-the-art for training-free length extrapolation, successfully generalizing models trained on a 4K context to 32 million tokens on RULER and BABILong. Our findings provide a clear and empirically-grounded set of design principles for developing future, highly-capable long-context language models.

## 1 Introduction

The recent proliferation of Large Language Models (LLMs) has catalyzed an urgent demand for models capable of processing long contexts (Brown et al., 2020; Touvron et al., 2023; OpenAI, 2024). However, the performance of standard Transformer architectures (Vaswani et al., 2023), which serve as the universal backbone, is well-documented to degrade sharply on out-of-distribution context lengths (Hsieh et al., 2024; Xiao et al., 2024; Lu et al., 2024; Kuratov et al., 2024). Concurrently, naively increasing the context window during training is often infeasible due to the quadratic complexity of the attention mechanism, which leads to prohibitive computational costs. This has spurred the community to investigate novel architectures designed explicitly for length extrapolation.

We posit that ideal length extrapolation necessitates two key properties: (1) the ability to train on

---

[*]Equal contribution
[†]Work done at Ant Group.
[‡]Corresponding authors

shorter sequences while achieving stable or even lower perplexity on longer sequences during inference, and (2) the capacity to effectively utilize the entire context. Significant research has focused on the first property. For instance, architectures like Sliding Window Attention (Beltagy et al., 2020) and recurrent models can maintain a nearly constant perplexity as the context extends beyond the training length (Buitrago & Gu, 2025). However, these methods fall short on the second property. The former is constrained by a fixed-size local receptive field, while the latter compresses the entire history into a fixed-size state, creating an information bottleneck. As revealed by in-context retrieval tasks like RULER(Hsieh et al., 2024), the inability to access and utilize arbitrary information from long contexts remains a critical limitation for many applications, from multi-round chatbots to agent-based systems (Liu et al., 2024; Pan et al., 2025; Guo et al., 2023).

Fortunately, recent methods based on retrieval and sparse attention (Mohtashami & Jaggi, 2023; Hu et al., 2025b;a) can maintain a near-constant perplexity on out-of-domain long sequences and achieve high accuracy on simple retrieval tasks like passkey retrieval. Their approach involves dividing the entire sequence into chunks, selecting the K most similar chunks and attending to the tokens within those chunks. However, on more complex retrieval tasks, their accuracy still degrades as context length increases. Moreover, a comprehensive analysis of why these models extrapolate so effectively—and what role their different designs play in this capability—is largely missing.

To address this gap, we present a systematic dissection of chunk-based sparse attention mechanisms, identifying the core architectural principles, theoretical motivations, and sparsity configurations that enable extreme generalization. Our main contributions are:

**1. A Unified Framework and Systematic Ablation.** We present a unified framework for chunk-based sparse attention and conduct a systematic ablation to identify the critical architectural components for generalization. We empirically demonstrate that an optimal combination of three design choices is essential: a non-linear **Chunk Encoder** for expressive representations, a dedicated **CLS token** for disentangling retrieval and content information, and a **Bypassing Residual Path** for stable information integration.

**2. Theoretical Motivation and Diagnostic Analysis.** We provide a theoretical motivation for the Chunk Encoder, framing it as a necessary non-linear approximator for full attention scores. We support our architectural claims with in-depth diagnostic analysis that correlates intermediate retrieval accuracy with final task performance, revealing *how* each component improves generalization by enhancing either retrieval prominence or information integration.

**3. State-of-the-Art Extrapolation and Insights on Sparsity.** By combining these optimal design principles, we establish a new state-of-the-art, generalizing a model trained on a 4K context to **32M tokens**. We further provide a detailed analysis on sparsity, showing that both a large context for contrastive learning and enforced selection sparsity during pre-training are crucial for generalization.

## 2 RELATED WORK

Approaches to long-context modeling can be broadly categorized. The first comprises local methods like Sliding Window Attention (Beltagy et al., 2020; Xiao et al., 2024). While effective at maintaining perplexity, they are fundamentally limited by a fixed-size local receptive field, preventing access to information beyond their local window. A second category includes recurrent architectures (Gu & Dao, 2024; Dao & Gu, 2020; Sun et al., 2023; Qin et al., 2023; Peng et al., 2023; Yang et al., 2024). Recent work shows that zero-initialized recurrent models struggle to generalize to longer sequences; it attributes the failure to models entering unexplored states and introduces State Passing and Truncated Backpropagation Through Time (Sutskever, 2013; Williams & Peng, 1990) to improve length generalization, achieving stable perplexity on out-of-distribution lengths (Buitrago & Gu, 2025). However, on challenging retrieval tasks (Hsieh et al., 2024), performance still drops significantly beyond the training length. This is because the entire context history is compressed into a fixed-size state, sacrificing the ability to recall specific, distant information losslessly. The third category seeks to extend full attention by modifying positional encodings (Press et al., 2022; Peng et al., 2024; Chen et al., 2023; Tan et al., 2025). Their training-free extrapolation gains are modest, typically limited to a small multiple of training length.

In pursuit of more scalable extrapolation, chunk-based sparse attention has emerged as a distinct and highly effective paradigm. Landmark Attention (Mohtashami & Jaggi, 2023) pioneered this by

using dedicated landmark tokens to represent chunks for retrieval. While innovative, its length extrapolation capability is limited to about 64x training length. NSA (Yuan et al., 2025) proposed an efficient hardware-aligned implementation of chunk selection but has very limited length extrapolation (Hu et al., 2025a). State-of-the-art training-free extrapolation has been achieved by Differentiable Retrieval-based Transformers (DRT) (Hu et al., 2025b) and RAMba (Hu et al., 2025a), which can generalize up to 1000x their training length on simple retrieval tasks by introducing a learnable hierarchical chunkwise sparse attention for retrieval. These methods also differ in their retrieval frequency, with DRT operating per-chunk while RAMba, NSA, and Landmark retrieve per-token. Our work builds upon the SWA+Hierarchical Sparse Attention (HSA) architecture with per-token retrieval, providing a systematic analysis to dissect and identify the specific architectural components responsible for the extreme generalization capabilities of this emerging paradigm.

## 3 PRELIMINARY

### 3.1 RANDOM CONTEXT ACCESS: THE KEY ABILITY TO UTILIZE FULL CONTEXT

Effective long-context reasoning hinges on the ability to flexibly retrieve information from any part of the preceding context. This capability, which we term *Random Context Access* (RCA), is critical for models to excel at tasks requiring the synthesis of distal information. We define it as follows:

***Definition:*** At any given step $t$, *Random Context Access* is the ability of a model to attend to any subset of tokens from the preceding context $\{x_0, \ldots, x_{t-1}\}$ in a data-dependent and lossless manner.

Under this definition, different architectures exist on a spectrum of capability. The standard Transformer, with its full-attention mechanism, perfectly exemplifies RCA by granting unrestricted access to the entire KV cache. In stark contrast, many efficient architectures trade this property for computational gains by imposing structural constraints. These include: **locality constraints**, in which models like Sliding Window Attention are confined to a local neighborhood; **information bottlenecks**, created by State Space Models (SSMs) that compress history into a lossy, fixed-size state; and **static access patterns** (e.g., Child et al. (2019); Beltagy et al. (2020)), where data-independent sparse attention predetermines information pathways, with more details in Apx. B.1. These architectural bottlenecks fundamentally limit a model's capacity for true long-range reasoning.

### 3.2 DYNAMIC CHUNKWISE SPARSE ATTENTION

Chunkwise sparse attention offers a practical and scalable approximation of RCA. The principle of dynamic chunkwise sparse attention mirrors an efficient strategy for an open-book exam. Rather than exhaustively reading the entire source material (akin to full attention), one first identifies key pages using high-level bookmarks (*landmarks*). Queries are then resolved by first consulting these landmarks to locate the relevant pages (*chunks*) and only then reading their detailed contents (*tokens*). This hierarchical retrieval drastically reduces the search space, with its success contingent on the quality of the landmarks.

Following this principle, our method provides access to the global context by dynamically selecting

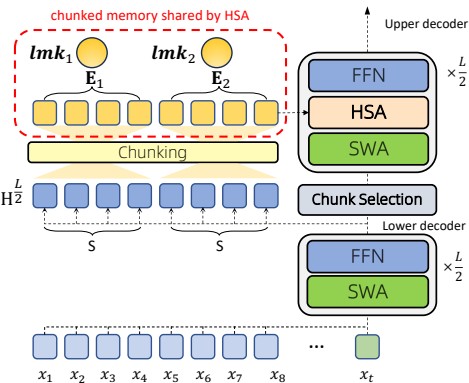

Figure 1: **SWA+HSA Architecture.** Our SWA+HSA model is primarily composed of Transformers with Sliding Window Attention (SWA), with layers divided into upper layers and lower layers. **Lower decoders** are composed of **SWA** followed by an FFN. At the architectural midpoint, a **chunking** layer splits representations from lower decoders into chunks, and encode them by chunk to build a global memory, each chunk of which contains a landmark vectors (**lmk**) and encoded chunks (**E**). **Upper decoders** each contains a local self-attention layer for local context, followed by an **HSA** to attend selected global memory and then an FFN.

a fixed number of chunks. While this imposes a chunkwise access pattern, it aligns with empirical findings that attention scores often exhibit a blockwise structure (Yuan et al., 2025) and that most queries only require a local subset of the context. This approach thereby offers a compelling trade-off between unrestricted context access and computational feasibility.

## 3.3 Hierarchical Sparse Attention and Model Design

As shown in Figure 1, our SWA+HSA model treats the global memory in a chunkwise approach. During decoding, the chunking layer gathers hidden representations into a buffer and encodes them as complete chunks. To maintain causality, the chunk containing the current token, as well as all subsequent chunks, are not accessible for retrieval. Following the notation of Sun et al. (2023), we use indices with $[\cdot]$ to indicate that it is indexed by chunk rather than by token. For instance, $\mathbf{H}_{[t]} := \mathbf{h}_{t \cdot C + 1 : (t+1) \cdot C} \in \mathbb{R}^{C \times d}$ is the chunked hidden states. At chunk selection layer, a query vector $\mathbf{q_t} \in \mathbb{R}^{d_{\text{Rtrv}}}$ computes a dot-product similarity score on landmark $\mathbf{lmk}_{[i]} \in \mathbb{R}^{d_{\text{Rtrv}}}$, formulated as $s_{t,i} = \mathbf{q_t} \cdot \mathbf{lmk}_{[i]} \in \mathbb{R}$. The indices of the top-$N$ chunks are then selected, forming set $\mathcal{I}_t$.

A critical design choice lies in how these selected chunks are weighted, as this determines how the retrieval signal is backpropagated. The weighting function $w_{t,i}$ is zero for all non-selected chunks. For the selected chunks, the weights are assigned according to distinct strategies for NSA (Eq. 1a), GCA (Eq. 1b), and HSA (Eq. 1c). Let $\mathcal{S}_t = \{s_{t,i}\}_{i \in \mathcal{I}_t}$.

$$w_{t,i} = 1 \quad \text{(1a)} \qquad w_{t,i} = \text{Softmax}(\mathcal{S}_t)_i \quad \text{(1b)} \qquad w_{t,i} = \text{StickBreak}(\mathcal{S}_t)_i \quad \text{(1c)}$$

This comparison highlights a clear progression: from the implicit binary weighting of NSA, to the proportional softmax weighting in GCA (Hu et al., 2025b), and finally to the recency-biased allocation via stick-breaking process[1] (Tan et al., 2025) used in HSA (our work and Hu et al., 2025a).

In the second stage, the selected chunk indices $\mathcal{I}_t$ and their corresponding weights $w_t$ are used to compute the final attention output. The HSA layer takes the query $\mathbf{Q}_t$ and the selected keys and values, $[\mathbf{K}_{[\mathcal{I}_t]}, \mathbf{V}_{[\mathcal{I}_t]}]$, and performs a weighted-sum over standard attention operations:

$$\mathbf{O_t} = \sum_{i \in \mathcal{I}_t} w_{t,i} \cdot \text{Attention}(\mathbf{Q}_t, \mathbf{K}_{[i]}, \mathbf{V}_{[i]}) = \sum_{i \in \mathcal{I}_t} w_{t,i} \cdot \text{Softmax}\left( \frac{\mathbf{Q}_t \mathbf{K}_{[i]}^T}{\sqrt{d_k}} \right) \mathbf{V}_{[i]} \qquad (2)$$

The direct participation of the weights $w_{t,i}$ in the forward computation (Eq. 2) creates a differentiable pathway where chunks deemed more important contribute more significantly to the final output. Crucially, in the backward pass, the gradients flowing to each chunk's parameters are scaled by their weight $w_{t,i}$. This enables the model to perform effective credit assignment, concentrating learning on the most relevant retrieved chunks.

## 4 Systematical study of retrieval and intra-chunk processing

### 4.1 Chunk Information preprocess

As shown in Fig. 1, $\mathbf{h}_t^{\frac{L}{2}}$ serves dual purposes, (1) passed to upper decoder layer as input, (2) gathered with adjacent hidden states to form the chunked hidden states $\mathbf{H}^{\frac{L}{2}}$ to generate global memory (chunked KV and landmarks). We define these operations through a pair of functions: $f(\cdot)$ for landmark aggregation and $g(\cdot)$ for intra-chunk processing. The final outputs are then derived via linear projections:

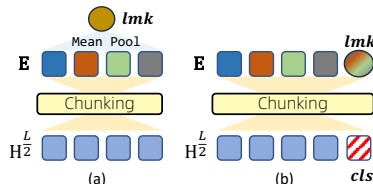

Figure 2: Design of Encoder: (a): Encoder w/o CLS (b): Encoder with a learnable CLS token.

---

[1]Specifically, let $\mathcal{I}'_t = \text{sort}(\mathcal{I}_t)$ be the selected chunk indices sorted by their scores, then $w_{t,k} = \sigma(s_{t,\mathcal{I}'_{t,k}}) \prod_{j<k}(1 - \sigma(s_{t,\mathcal{I}'_{t,j}}))$ where $\sigma$ is the sigmoid function.

Table 1: **Unified Comparison of Chunk Processing:** The Encoder is a bidirectional Transformer encoder shared by $f(\cdot)$ and $g(\cdot)$. For the "w/ CLS" variant, $[0]$ selects the output corresponding to the prepended $\mathbf{x}_{\text{CLS}}$ token, and $[1:]$ selects the outputs for the original chunk hidden states $\mathbf{H}_{[i]}$.

| Configuration | $f(\mathbf{H}_{[i]})$ | $g(\mathbf{H}_{[i]})$ |
|---|---|---|
| NSA | MeanPool($\mathbf{H}_{[i]}$) | $\mathbf{H}_{[i]}$ |
| HSA w/o Encoder | MeanPool(Norm($\mathbf{H}_{[i]}$)) | RMSNorm($\mathbf{H}_{[i]}$) |
| HSA w/ Encoder w/o CLS | MeanPool(Encoder($\mathbf{H}_{[i]}$)) | Encoder($\mathbf{H}_{[i]}$) |
| HSA w/ Encoder w/ CLS | Encoder($[\mathbf{x}_{\text{CLS}}; \mathbf{H}_{[i]}]$)$[0]$ | Encoder($[\mathbf{x}_{\text{CLS}}; \mathbf{H}_{[i]}]$)$[1:]$ |

$$\mathbf{lmk}_{[i]} = \text{Linear}(f(\mathbf{H}_{[i]})) \in \mathbb{R}^{d_{\text{Rtrv}}} \quad \text{and} \quad [\mathbf{K}_{[i]}, \mathbf{V}_{[i]}] = \text{Linear}(g(\mathbf{H}_{[i]})) \in \mathbb{R}^{h_{kv} \times (d_k + d_v)}$$

The different architectural configurations we investigate, summarized in Table 1, can be expressed as joint definitions of $(f, g)$. In the "w/ CLS" variant, we prepend a learnable token, $\mathbf{x}_{\text{CLS}}$, to the input chunk $\mathbf{H}_{[i]}$, as shown in Fig. 2. The Encoder processes this combined sequence, and its output corresponding to the $\mathbf{x}_{\text{CLS}}$ position is used to form the landmark, while the remaining outputs form the KV cache.

**Chunked Hidden States: Disentangling Representations for Retrieval and Prediction.** The hidden state $\mathbf{h}_t^{\frac{L}{2}}$ must be immediately useful for autoregressive next token prediction, while also serving as a durable memory for future retrieval. Considering the dual purposes of $\mathbf{h}_t^{\frac{L}{2}}$, we posit that simply operating on these raw hidden states conflates these two distinct functional objectives. The introduction of non-linear layers aims to disentangle these two roles. By processing the raw states, these components can produce refined representations specifically optimized for retrieval, separate from the immediate demands of next-token prediction. A RMSNorm provides a simple mechanism to introduce non-linearity, while the Encoder takes this a step further by explicitly learning to re-encode the chunk's information.

**Landmark: Approximating Full Attention via Learnable Summarization.** We frame HSA as a structured approximation of full attention. The goal is for the HSA weight $\hat{\alpha}_{t,i}$(Eq. 3b) to be a faithful approximation of the full attention weight $\alpha_{t,i}$ (Eq. 3a).

$$\alpha_{t,i} = \frac{\exp(\mathbf{q}_t \cdot \mathbf{k}_i)}{\sum_{j=1}^{t-1} \exp(\mathbf{q}_t \cdot \mathbf{k}_j)} \quad (3a) \qquad \hat{\alpha}_{t,i} = w_{t,c_i} \cdot \frac{\exp(\mathbf{q}_t \cdot \mathbf{k}_i)}{\sum_{j \in c_i} \exp(\mathbf{q}_t \cdot \mathbf{k}_j)} \quad (3b)$$

Formally, the ideal chunk weight $w_{t,c_i}$ should approximate the ratio of its chunk's attention mass to the global attention mass, as shown in Eq. 4a. However, since a landmark is generated using only local chunk information, it is impossible to compute the global denominator. Therefore, the model must learn a practical proxy: the unnormalized selection score $s_{t,c_i}$ must be made proportional to the chunk's unnormalized attention mass, a relationship defined in Eq. 4b.

$$w_{t,c_i} \approx \frac{\sum_{j \in c_i} \exp(\mathbf{q}_t \cdot \mathbf{k}_j)}{\sum_{k=1}^{t-1} \exp(\mathbf{q}_t \cdot \mathbf{k}_k)} \quad (4a) \qquad \mathbf{q}_t \cdot \mathbf{lmk}_{[i]} \propto \sum_{j \in c_i} \exp(\mathbf{q}_t \cdot \mathbf{k}_j) \quad (4b)$$

The core challenge is thus to generate a landmark $\mathbf{lmk}_{[i]}$ that satisfies the relationship in Eq. 4b. The right-hand side of this equation is a highly nonlinear function of the keys within the chunk. A simple linear operator like MeanPool is insufficient for this task. A multilayer Chunk Encoder, as a powerful learnable function, is essential for learning this complex relationship. Adding a CLS token and using its output as landmark, further improve the nonlinearity and learnability compared with mean-pooling of the Chunk Encoder outputs.

## 4.2 SKIP CONNECTION DESIGN FOR CROSS-LAYER INFORMATION FUSION

Some HSA layers in our model can retrieve information to the shared long-range memory, by attending to the key-value states at the output of the chunking layer. This design introduces a challenge: how to effectively fuse the more abstract information of the current upper layer, $\mathbf{x}_{\text{in}}$, with the more

literal, lower-level context retrieved by the HSA module, $\mathcal{H}(\mathbf{x}_{\text{in}})$.

We investigate two alternative skip connection strategies for integrating the HSA layer within the upper decoder blocks. In both designs, the output of the HSA layer is added to the input residual to form an intermediate representation, $\mathbf{x}'$ (Eq. 5). This representation is then processed by the subsequent MLP block, $\mathcal{M}(\cdot)$. The key distinction between the two strategies lies in the final residual connection. The **Standard Sequential Path** (Eq. 6a) follows the conventional Transformer design, adding the MLP's output back to the intermediate representation $\mathbf{x}'$. In contrast, the **Bypassing Residual Path** (Eq. 6b) provides a more controlled mechanism, ensuring the direct output from the HSA module *bypasses* the final residual addition.

$$\mathbf{x}' = \mathbf{x}_{\text{in}} + \mathcal{H}(\mathbf{x}_{\text{in}}) \quad (5)$$

$$\mathbf{x}_{\text{out}} = \mathbf{x}' + \mathcal{M}(\mathbf{x}') = \mathbf{x}_{\text{in}} + \mathcal{M}(\mathbf{x}') + \mathcal{H}(\mathbf{x}_{\text{in}}) \quad (6a)$$

$$\mathbf{x}_{\text{out}} = \mathbf{x}_{\text{in}} + \mathcal{M}(\mathbf{x}') \quad (6b)$$

We hypothesize that the Bypassing Residual Path is particularly suited for our cross-layer attention architecture. The direct addition of cross-layer information $\mathcal{H}(\mathbf{x}_{\text{in}})$ to the highly-processed residual stream, as in the Standard Sequential Path may disrupt the information flow. On the contrary, the Bypassing Residual Path tasks the MLP with explicitly learning how to resolve the information gap between network depths.

## 5 EXPERIMENTS

We organize our experiments as follows. Section 5.1 describes the experimental setup, including model architecture, training details, benchmarks, and baselines. Section 5.2 presents length extrapolation results on BabiLong and RULER, demonstrating the superiority of our SWA+HSA architecture. Section 5.3 provides an in-depth analysis of the retrieval mechanism, disentangling retrieval accuracy from information integration. Section 5.4 investigates the role of sparsity in enabling extreme-length generalization. Finally, Section 5.5 validates the scalability and general capabilities of our approach.

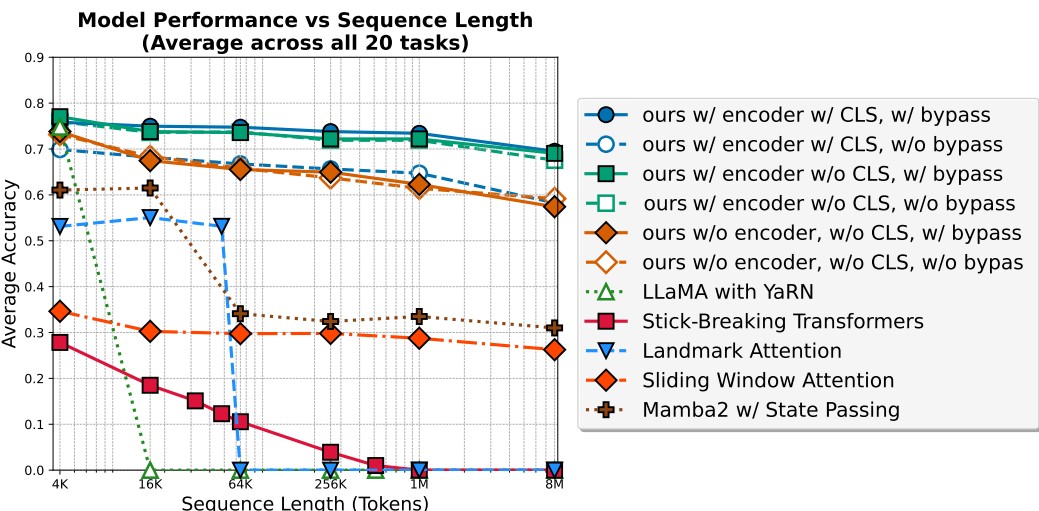

Figure 3: **BabiLong Evaluation Results:** The best SWA+HSA model *(ours)* demonstrates robust long-context extrapolation, maintaining high accuracy up to 8M tokens. In stark contrast, baseline models exhibit clear limitations. Full-attention methods collapse to near-zero accuracy shortly after their training length. Prior sparse methods like Landmark Attention show better generalization but also fail decisively at longer contexts ($\geq$64k). Sliding Window Attention plateaus at a low accuracy consistent with random guessing, while Mamba2's performance degrades substantially.

Table 2: **RULER evaluation results.** Our best SWA+HSA model maintains high accuracy up to 32M tokens, significantly outperforming all baselines in long-context scenarios. In Model Specs, 'Enc' denotes the number of encoder layers, 'CLS' indicates the use of a CLS token, and 'Bypass' refers to the Bypassing Residual Path.

| Model | Enc | CLS | Bypass | S-N | MQ-N | VT | Avg. | S-N | MQ-N | VT | Avg. | S-N | MQ-N | VT | Avg. |
|---|---|---|---|---|---|---|---|---|---|---|---|---|---|---|---|
| | Model Specs | | | ctx-len=4K | | | | ctx-len=32K | | | | ctx-len=128K | | | |
| Mamba2 | 0 | — | — | **99.54** | 15.96 | 80.80 | 65.43 | 1.48 | 1.49 | 0.40 | 1.12 | 0.00 | 0.00 | 0.00 | 0.00 |
| NSA | 0 | — | — | 22.82 | 3.15 | 12.43 | 12.80 | 0.00 | 0.00 | 4.48 | 1.49 | 0.00 | 0.00 | 0.00 | 0.00 |
| Llama-Yarn | 0 | — | — | 92.39 | 69.11 | 93.51 | 85.00 | 0.00 | 0.00 | 0.00 | 0.00 | 0.00 | 0.00 | 0.00 | 0.00 |
| Stick-Break | 0 | — | — | 76.72 | 22.17 | 37.20 | 45.36 | 73.88 | 24.63 | 31.34 | 43.28 | 33.33 | 18.18 | 30.30 | 27.27 |
| Landmark | 0 | — | — | 83.00 | 37.00 | 40.00 | 53.33 | 83.00 | 15.00 | 20.00 | 39.33 | 0.00 | 0.00 | 0.00 | 0.00 |
| SWA+HSA | 0 | no | no | 84.40 | 61.30 | 37.02 | 60.91 | 78.97 | 38.80 | 12.14 | 43.30 | 77.49 | 35.14 | 7.80 | 40.14 |
| SWA+HSA | 1 | no | no | 81.20 | 68.40 | 87.80 | 79.13 | 81.00 | 66.20 | 87.40 | 78.20 | 77.20 | 62.60 | 86.60 | 75.47 |
| SWA+HSA | 1 | yes | no | 79.80 | 70.40 | 90.00 | 80.07 | 81.00 | 65.40 | 86.40 | 77.60 | 75.60 | 62.20 | 84.80 | 74.20 |
| SWA+HSA | 2 | no | no | 77.39 | 67.52 | 89.24 | 78.05 | 78.58 | 65.75 | 87.66 | 77.33 | 74.83 | 64.76 | 87.27 | 75.62 |
| SWA+HSA | 2 | yes | no | 86.28 | 71.96 | **93.98** | 84.07 | 87.46 | 68.11 | **92.89** | 82.82 | 85.69 | 67.32 | **93.48** | 82.16 |
| SWA+HSA | 0 | no | yes | 90.33 | 71.77 | 72.26 | 78.12 | 88.06 | 71.08 | 69.20 | 76.11 | 84.21 | 69.00 | 65.75 | 72.99 |
| SWA+HSA | 1 | no | yes | 89.20 | 77.00 | 93.80 | 86.67 | 89.40 | 75.60 | 90.80 | 85.27 | 84.40 | 73.40 | 90.20 | 82.67 |
| SWA+HSA | 1 | yes | yes | 91.20 | 75.20 | 88.80 | 85.07 | 89.20 | 72.80 | 89.80 | 83.93 | 88.20 | 74.00 | 87.80 | 83.33 |
| SWA+HSA | 2 | no | yes | 91.71 | 72.56 | 93.68 | 85.98 | 88.15 | 72.36 | 92.20 | 84.24 | 86.97 | 71.17 | 92.99 | 83.71 |
| SWA+HSA | 2 | yes | yes | 91.51 | **78.78** | 93.48 | **87.92** | 89.14 | 73.84 | 91.61 | 84.86 | 88.94 | 76.21 | 91.81 | **85.65** |
| Models | Enc | CLS | Bypass | S-N | MQ-N | VT | Avg. | S-N | MQ-N | VT | Avg. | S-N | MQ-N | VT | Avg. |
| | Model Specs | | | ctx-len=1M | | | | ctx-len=8M | | | | ctx-len=32M | | | |
| SWA+HSA | 0 | no | no | 72.28 | 21.78 | 3.96 | 32.67 | 61.39 | 6.93 | 1.98 | 23.43 | 61.80 | 1.12 | 0.00 | 20.97 |
| SWA+HSA | 1 | no | no | 72.73 | 58.18 | 72.73 | 67.88 | 69.09 | 34.55 | 58.18 | 53.94 | 70.91 | 47.27 | 43.64 | 53.94 |
| SWA+HSA | 1 | yes | no | 63.64 | 40.00 | 54.55 | 52.73 | 69.09 | 38.18 | 49.09 | 52.12 | 54.55 | 21.82 | 30.91 | 35.76 |
| SWA+HSA | 2 | no | no | 63.37 | 63.37 | 72.28 | 66.34 | 58.42 | 48.51 | 66.34 | 57.76 | 67.42 | 42.70 | 70.79 | 60.30 |
| SWA+HSA | 2 | yes | no | 83.17 | 46.53 | 82.18 | 70.63 | 78.22 | 28.71 | 80.20 | 62.38 | 82.02 | 24.72 | 61.80 | 56.18 |
| SWA+HSA | 0 | no | yes | 80.20 | 64.36 | 45.54 | 63.37 | 71.29 | 46.53 | 15.84 | 44.55 | 82.02 | 31.46 | 3.37 | 38.95 |
| SWA+HSA | 1 | no | yes | 69.09 | 56.36 | 45.45 | 56.97 | 58.18 | 30.91 | 10.91 | 33.33 | 63.64 | 36.36 | 12.73 | 37.58 |
| SWA+HSA | 1 | yes | yes | **89.09** | 69.09 | 80.00 | 79.39 | 72.73 | 78.18 | 76.36 | 75.76 | 85.45 | 69.09 | 69.09 | 74.54 |
| SWA+HSA | 2 | no | yes | 85.15 | 74.26 | 74.26 | 77.89 | 78.22 | 65.35 | 65.35 | 69.64 | 82.02 | 66.29 | 41.57 | 63.29 |
| SWA+HSA | 2 | yes | yes | 88.12 | **76.24** | **93.07** | **85.81** | 79.21 | 77.23 | 90.10 | 82.18 | 82.02 | 68.54 | **88.76** | **79.77** |

## 5.1 EXPERIMENTAL SETUP

Our experiments are conducted on a series of models based on the SWA+HSA architecture detailed in Figure 1. All model variants contain approximately 240M parameters and are trained with a context length of 4K. Unlike the original DRT (Hu et al., 2025b) which uses Grouped Cross Attention (GCA) for per-chunk retrieval, our architecture employs HSA with a stick-breaking weighting strategy (Eq. 1c) to enable per-token retrieval, providing greater flexibility for dynamic access to long-range information.

Key hyperparameters for this setup are fixed across experiments. The sliding window size for the lower layers is 512 tokens, and the HSA chunk size is 64. Unless otherwise noted, we use a Top-K of 8 for chunk selection, giving the HSA module an effective receptive field of 512 tokens. A comprehensive description of our training recipes and further hyperparameter details is provided in Appendix C.

**Benchmark.** We evaluate our models on two long-context benchmarks: **BabiLong** (Kuratov et al., 2024) and **RULER** (Hsieh et al., 2024). BabiLong is designed to simulate complex question-answering scenarios over long documents. It extends the classic bAbI dataset (Weston et al., 2015) by embedding the original short stories and factual sentences into a long distractor context. This design challenges a model to first perform robust, long-range information retrieval to locate all relevant facts, and then apply the reasoning required by the original task. Consequently, BabiLong tests retrieval and reasoning capabilities simultaneously.

To complement this and more directly assess a model's core retrieval ability in isolation, we also use RULER. From RULER, we select three representative sub-tasks: Single-Needle (**S-N**); Multi-Query Needle (**MQ-N**), where 2 of 6 key-value pairs are queried; and Variable Tracking (**VT**), which tests transitive assignment chains. Collectively, these tasks provide a targeted evaluation of a model's capacity for precise, long-range information retrieval and tracking simple logical chains. As RULER requires accurately retrieving several tokens, it provides a strict test of models' random context access ability.

**Baselines.** We compare our method against several baselines. Our primary full-attention baseline is Llama with YaRN and Mamba2 with State Passing (Buitrago & Gu, 2025), which is trained

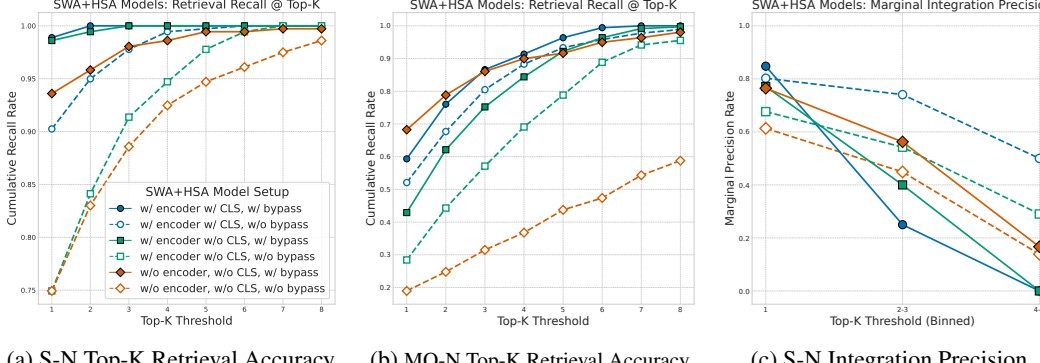

(a) S-N Top-K Retrieval Accuracy    (b) MQ-N Top-K Retrieval Accuracy    (c) S-N Integration Precision

Figure 4: **Analysis of Retrieval and Utilization Mechanisms.** Subplots (a) and (b) show retrieval accuracy (Recall@Top-K) on the S-N and MQ-N tasks, respectively. The results highlight that the Bypass mechanism is the most critical component for successful retrieval, consistently outperforming variants without it. The learnable encoder and CLS token also provide noticeable improvements. Subplot (c) plots the conditional answer accuracy on the S-N task, given that the correct chunk was retrieved at a specific rank (by weight). Across all models, integration precision drops sharply as the rank of the correct chunk decreases. This demonstrates a stricter requirement: a chunk must not only be successfully retrieved (recall) but also be assigned a high-ranking weight to be effectively utilized by the model.

on a 4k context length. To ensure a fair comparison with other efficient attention mechanisms, we include Landmark Attention, configured to match our model's training FLOPs during training by using a sequence length of 756. Furthermore, to benchmark against methods with a similar computational budget, we include Stick-breaking Attention and Sliding Window Attention. Both of these are trained with a 1k context window to keep the receptive field same to our approach.

## 5.2 Length Extrapolation Results

As shown in Figure 3, the results on BabiLong highlight the stark differences in extrapolation capability across architectures. The benchmark's classification-like nature establishes a random-guess baseline around 0.3 accuracy, a level to which Sliding Window Attention predictably converges due to its limited context. Other baselines also fail to generalize, with full-attention and Landmark Attention collapsing at various points beyond their training regimes. Crucially, the figure also reveals the importance of our key design choices within the SWA+HSA architecture. The model variants with a learnable chunk encoder consistently outperform those without.

To further validate these findings and provide a more granular analysis of retrieval performance, we turn to the RULER benchmark, with detailed results in Table 2. The trends observed here strongly corroborate our conclusions from BabiLong. First, comparing our SWA+HSA models against the baselines reveals a significant performance gap in extrapolation. Llama with Yarn (Llama-Yarn), despite its strong in-domain performance at 4K, fails catastrophically at 32K, with their scores dropping to zero. Stick-breaking Attention (Stick-Break) and Landmark Attention demonstrate some extrapolation ability but exhibit significant performance decay and are consistently outperformed by our models on longer contexts.

Second, while all our model variants showcase robust long-context performance, the results also highlight a clear performance hierarchy among our different architectural configurations. Our full model (Enc=2, CLS=yes, Bypass=yes) consistently achieves state-of-the-art results. It not only secures the highest average score of 87.92% at the 4K training length but also degrades remarkably gracefully, maintaining an average score of 79.77% even at an extreme context length of 32M. This indicates that the specific combination of our proposed components is critical for unlocking maximum performance, an interaction we will analyze in detail in the subsequent section.

Table 3: **Ablation study on sparsity using the RULER benchmark.** The interplay between training context length and sparsity is critical for extreme extrapolation. A model trained on a 4K context with high selection sparsity (Top-K=8) ultimately achieves the best performance at 32M tokens.

| Model | Model Specs | | | ctx-len=4K | | | | ctx-len=32K | | | | ctx-len=128K | | | |
|---|---|---|---|---|---|---|---|---|---|---|---|---|---|---|---|
| | TPR | Top-K | Train-Length | S-N | MQ-N | VT | Avg. | S-N | MQ-N | VT | Avg. | S-N | MQ-N | VT | Avg. |
| SWA+HSA | 1 | 8 | 1024 | 38.68 | 2.50 | 11.60 | 17.59 | 51.49 | 0.75 | 0.00 | 17.41 | 36.36 | 0.00 | 0.00 | 12.12 |
| SWA+HSA | 1 | 8 | 2048 | 95.27 | 67.90 | 56.86 | 73.34 | 96.27 | 71.64 | 10.45 | 59.45 | 87.88 | 63.64 | 18.18 | 56.57 |
| SWA+HSA | 1 | 8 | 4096 | 91.51 | 78.78 | 93.48 | 87.92 | 89.14 | 73.84 | 91.61 | 84.86 | 88.94 | 76.21 | 91.81 | 85.65 |
| SWA+HSA | 1 | 8 | 8192 | 98.70 | 90.54 | 96.75 | 95.33 | 95.52 | 92.54 | 97.01 | 95.02 | 96.97 | 87.88 | 96.97 | 93.94 |
| SWA+HSA | 1 | 64 | 4096 | 94.97 | 88.85 | 97.73 | 93.85 | 95.56 | 86.38 | 97.63 | 93.19 | 91.51 | 87.96 | 98.72 | 92.73 |
| SWA+HSA | 8 | 8 | 4096 | 93.69 | 31.82 | 89.89 | 71.80 | 93.28 | 24.63 | 88.06 | 68.66 | 93.94 | 6.06 | 63.64 | 54.55 |

| Model | Model Specs | | | ctx-len=1M | | | | ctx-len=8M | | | | ctx-len=32M | | | |
|---|---|---|---|---|---|---|---|---|---|---|---|---|---|---|---|
| | TPR | Top-K | Train-Length | S-N | MQ-N | VT | Avg. | S-N | MQ-N | VT | Avg. | S-N | MQ-N | VT | Avg. |
| SWA+HSA | 1 | 8 | 4096 | 88.12 | 76.24 | 93.07 | 85.81 | 79.21 | 77.23 | 90.10 | 82.18 | 82.02 | 68.54 | 88.76 | 79.77 |
| SWA+HSA | 1 | 8 | 8192 | 97.50 | 84.00 | 87.00 | 89.50 | 97.50 | 75.00 | 65.00 | 79.17 | 95.00 | 45.00 | 45.00 | 61.67 |
| SWA+HSA | 1 | 64 | 4096 | 78.18 | 90.91 | 96.36 | 88.48 | 51.31 | 76.36 | 94.55 | 74.07 | 50.91 | 69.09 | 89.09 | 69.70 |

## 5.3 IN-DEPTH ANALYSIS OF RULER RETRIEVAL

To further understand the mechanisms behind our architectural designs, we move beyond final task scores to conduct a diagnostic analysis of the intermediate retrieval process. Specifically, we investigate the relationship between the accuracy of chunk selection and the correctness of the final model output on the **S-N** and **MQ-N** tasks. In these retrieval-centric tasks, we can unambiguously identify the critical "needle" chunk(s) containing the information required to answer the prompt. We then measure the retrieval recall for these essential chunks and correlate it with final answer accuracy. This allows us to disentangle two primary failure modes: a failure to *retrieve* the relevant context versus a failure to *utilize* the context once it has been retrieved. Given that the evaluation context length far exceeds the local sliding window, successful task performance becomes contingent on accurate long-range retrieval in most cases.

Our analysis reveals that even at an extreme context length of 8M, most model configurations, with the notable exception of the poorly performing baseline (without an encoder or bypass path), successfully retrieve the correct "needle" chunk within their top-k(k=8) selections. The significant performance gap, therefore, does not originate from a complete failure of retrieval, but rather from two more subtle factors: **Retrieval Prominence:** Superior models assign a significantly higher rank and selection weight to the correct chunk within the top-k set, indicating a more precise and confident retrieval score, as indicated in Fig. 4a and Fig. 4b. As shown in Fig. 4c, the model can integrate information more accurately if the target information is assigned higher importance among the selected chunks. **Information Integration:** After successfully retrieving the relevant chunk(s), better-performing models are generally more effective at integrating this information to produce the correct final answer.

Our experiments empirically validate the role of our proposed architectural components in improving both these aspects: **1. The Bypassing Residual Path is crucial for effective information integration.** We observe that the bypassing path ('Bypass=yes') consistently and significantly improves final answer accuracy, even when the correct chunks are among selected chunks. This provides strong evidence for our hypothesis that retrieved information requires dedicated modulation before being integrated into the main residual stream. The bypassing design proves to be a highly effective mechanism for this modulation. **2. The Chunk Encoder and CLS token improve retrieval prominence.** Our results show a direct positive correlation between using the encoder and CLS token and the rank of the correct chunk. These models have the same total parameter count; Models with one more encoder layer have one less lower-decoder layer. The fact that this architectural trade-off leads to superior performance on retrieval-intensive tasks demonstrates that the encoder is a more parameter-efficient mechanism for learning robust retrieval representations and adding a CLS token to generate landmark can improve the parameter efficiency of the chunk encoders. Based on these findings, we also significantly improve length generalization of RAMba, detailed in Appendix. D.

## 5.4 THE ROLE OF SPARSITY

The effectiveness of our model critically depends on the careful orchestration of sparsity across multiple dimensions. We conduct a systematic ablation study to analyze three key dimensions of

sparsity using our best-performing configuration, with detailed results presented in Table 3.

**Sufficient Training Length.**  This dimension specifies the context window size used during training, which determines the pool of chunk candidates available for the sparse attention mechanism to learn from. A longer training context exposes the model to more diverse retrieval scenarios, enabling it to develop a more robust retrieval policy through richer contrastive signals. The empirical evidence strongly supports this: models trained on 4K contexts successfully extrapolate up to $8000\times$ their training length, while those trained on only 1K contexts fail to generalize beyond the training regime. Further extending training length to 8K brings additional gains, confirming a clear positive correlation between training context length and extrapolation capability.

**Selection Sparsity (Top-K).**  This parameter controls the number of chunks the model can attend to at each decoding step, directly regulating the sparsity of the retrieval mechanism. While a large Top-K value (e.g., 64) permits nearly dense global attention, a small Top-K (e.g., 8) enforces strict sparsity, compelling the model to learn highly discriminative retrieval. This enforced sparsity during training proves essential for extreme-length generalization. At 32M tokens, the model trained with Top-K=8 substantially outperforms its Top-K=64 counterpart, demonstrating that a sparse retrieval policy learned during training generalizes more effectively to the resource constraints and scale demands of inference at unprecedented sequence lengths.

**Retrieval Frequency (Tokens Per Retrieval).**  This dimension defines the retrieval interval: how many tokens are generated before the HSA mechanism refreshes its selected chunks. In other words, it specifies whether chunk selection is updated every single token (per-token retrieval) or once per block of multiple tokens (per-chunk retrieval). Two competing strategies exist: *per-token retrieval* offers maximal flexibility through fine-grained, dynamic context access at every step, while *per-chunk retrieval* potentially provides more stable training signals by reducing update frequency. To empirically determine the optimal strategy, we compare per-token retrieval against per-8-token retrieval. The results show a clear tradeoff: per-8-token retrieval achieves marginal gains on simple single-needle tasks (S-N) but suffers severe degradation on complex multi-query tasks (MQ-N, VT) requiring multiple targeted retrievals. This decisively favors per-token retrieval as the more robust choice for our final architecture.

## 5.5 Scalability and General capabilities

To further validate the scalability and general capabilities of our proposed architecture, we evaluate a scaled-up Mixture-of-Experts (MoE) variant (8B parameters, 1B active). The detailed training recipe and model specifications follow the setup in Hu et al. (2025c). As shown in Appendix A, our architecture achieves competitive performance on general reasoning benchmarks (e.g., GSM8K, MMLU) compared to standard Transformers, confirming that our length-extrapolation modifications do not compromise short-context capabilities.

## 6 Conclusion

In this work, we conducted a systematic investigation into the architectural principles underpinning extreme length generalization in chunk-based sparse attention models. Through a unified framework and comprehensive ablation studies, we demonstrated that a combination of three design principles is critical: (1) an expressive, non-linear Chunk Encoder with a dedicated CLS token to generate disentangled representations for retrieval and content; (2) a Bypassing Residual Path to stably integrate retrieved global information without disrupting the local processing stream; and (3) enforced selection sparsity during pre-training to align the model's learned retrieval policy with the demands of inference on unseen, longer contexts. Our in-depth analysis revealed precisely how these components improve performance by enhancing either retrieval prominence or information integration. By combining these principles, we developed a model that generalizes from a 4K training context to 32 million tokens without fine-tuning. Ultimately, our findings distill the complex design space into a clear, empirically-grounded set of guidelines for developing future, highly-capable long-context language models.

## ACKNOWLEDGEMENTS

We thank the anonymous reviewers for their insightful comments. This work was supported by the Ant Group Research Intern Program.

## ETHICS STATEMENT

The authors have read and adhere to the ICLR Code of Ethics. This work is foundational, focusing on the architectural principles of language models rather than a specific application. To uphold the principles of scientific transparency and reproducibility, our experiments were conducted exclusively on the publicly available datasets, and we commit to releasing our source code upon de-anonymization to allow for verification and future research. We believe that transparent and reproducible research is crucial for the responsible development of future AI systems.

## REPRODUCIBILITY STATEMENT

To ensure the reproducibility of our findings, we have provided comprehensive details of our experimental setup. The proposed architectural modifications are described thoroughly, with a detailed architectural diagram and conceptual overview in Figure 1. All hyperparameters and training procedures, including the specific recipes for pre-training and supervised fine-tuning on the RULER and BabiLong benchmarks, are thoroughly documented in Appendix C. The datasets used are publicly available, and our evaluation setup is described in Section 5.1. Furthermore, we commit to releasing our source code upon de-anonymization to allow for full verification of our results and to support future research in this area.

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

APPENDIX

## A    EVALUATION ON GENERAL CAPABILITIES AT SCALE

A critical concern for specialized long-context architectures is whether the modifications introduced to support extreme length extrapolation—specifically the hierarchical sparse attention and chunk encoding—negatively impact performance on standard short-context tasks.

To empirically verify the scalability of our approach and ensure no performance regression occurs in general domains, we evaluate a scaled-up version of our model, denoted as **SWA-HSA-MoE**.

**Experimental Setup.**    We conduct a controlled comparison between our method and a standard Transformer baseline in the large-scale regime:

- **Model Scale:** Both models are 8B-parameter Mixture-of-Experts (MoE) architectures with approximately 1B active parameters per token.
- **Training Data:** Both models were pre-trained on 8 trillion tokens using identical data mixtures and training recipes.
- **Architecture:** The **Transformer-MoE** baseline uses standard dense attention with MoE Feed-Forward Networks (FFNs). The **SWA-HSA-MoE** replaces the attention mechanism with our proposed Hierarchical Sparse Attention (HSA) framework (including the non-linear chunk encoder and bypassing residual path) while retaining the identical MoE FFN structure.

We refer readers to Hu et al. (2025c) for comprehensive details regarding the training infrastructure, MoE implementation, and dataset composition used in these experiments.

**Results.**    We evaluate both models on a diverse set of standard short-context benchmarks covering knowledge, reasoning, math, and coding. As shown in Table 4, **SWA-HSA-MoE** achieves competitive or superior performance across the majority of tasks:

1. **No Regression:** The overall average score of SWA-HSA-MoE (57.69) surpasses the standard Transformer-MoE baseline (55.62), confirming that our long-context mechanisms do not compromise core language modeling capabilities.

2. **Reasoning & Coding Gains:** Notably, our model shows significant improvements in complex reasoning tasks, including GSM8K (+0.61), MATH (+4.02), and coding benchmarks like MBPP+ (+5.56). This suggests that the hierarchical selection capability may offer benefits even in non-retrieval-heavy contexts by efficiently filtering information.

Table 4:    **General Capabilities Evaluation (8B MoE).** Comparison between a standard Transformer-MoE and our proposed SWA-HSA-MoE. Both models have 8B total parameters (1B active) and are trained on 8T tokens. Our method demonstrates robust performance on standard short-text benchmarks, outperforming the baseline on 6 out of 7 tasks.

| Benchmark | Category | Transformer-MoE | SWA-HSA-MoE (Ours) |
|---|---|---|---|
| MMLU | Knowledge & Reasoning | **58.74** | 57.83 |
| PIQA | Knowledge & Reasoning | 77.48 | **78.84** |
| BBH | Knowledge & Reasoning | 50.34 | **51.70** |
| GSM8K | Math Reasoning | 66.41 | **67.02** |
| MATH | Math Reasoning | 37.96 | **41.98** |
| HumanEval+ | Coding | 48.17 | **50.61** |
| MBPP+ | Coding | 50.26 | **55.82** |
| **Average** | **Overall** | 55.62 | **57.69** |

These results serve as strong empirical evidence that the architectural principles proposed in this

work—specifically the synergy between the chunk encoder and bypassing residual path—are scalable and universally effective, extending context length significantly without sacrificing the model's general utility.

## B ADDITIONAL RELATED WORK

In the main body of our paper, we focus on dynamic sparse attention methods specifically designed for length extrapolation. Here, we provide a broader overview of the rich literature on sparse attention, which has historically focused on a different goal: **reducing the quadratic complexity of attention for long, but still in-distribution, sequences**. We categorize these methods into static and dynamic approaches, and for each, we explain why their design principles do not typically lend themselves to training-free length extrapolation.

### B.1 STATIC SPARSE ATTENTION

Static sparse attention methods reduce computational complexity by restricting each token to a predefined, fixed subset of other tokens. These data-independent patterns are highly efficient but lack the flexibility required for length extrapolation. Early work like Sparse Transformer (Child et al., 2019) introduced fixed strided and dilated patterns, a foundational concept that influenced many subsequent models. This idea evolved into more complex topologies that combine local windowed attention with some form of global connectivity. For instance, Longformer (Beltagy et al., 2020) and ETC (Ainslie et al., 2020) designated a few "global" tokens to act as information hubs for the entire sequence. BigBird (Zaheer et al., 2020) generalized this by combining local, global, and random connections to approximate the small-world properties of a dense graph. These methods, while successfully extending the manageable context length for tasks like long-document processing, are fundamentally tied to the positional relationships learned during training. Their fixed patterns and reliance on learned positional biases prevent them from generalizing to sequence lengths orders of magnitude beyond what they were trained on.

### B.2 DYNAMIC SPARSE ATTENTION

Unlike static methods, dynamic sparse attention mechanisms determine attention patterns adaptively based on the input content, aiming to approximate the expressiveness of full attention by focusing computation on relevant tokens. Early approaches often relied on heuristic-based grouping or clustering. Reformer (Kitaev et al., 2020) pioneered the use of locality-sensitive hashing (LSH) to bucket tokens for attention, while the Routing Transformer (Roy et al., 2021) used online k-means clustering. These methods were a significant step towards content-aware sparsity. Another line of work focused on memory, with Transformer-XL (Dai et al., 2019) introducing a recurrent memory and Memorizing Transformers (Wu et al., 2022) using a kNN index for retrieval from an external long-term memory. While these methods are "dynamic", their selection mechanisms are still parameterized and optimized for the context lengths seen during training. The learned LSH functions, cluster centroids, or memory management policies are not inherently length-agnostic and thus do not extrapolate well. More recent hardware-aware methods like NSA (Yuan et al., 2025) and MoBA (Lu et al., 2025) have achieved significant practical speedups, but their design goal remains computational efficiency within a known context distribution, not training-free generalization to extreme lengths. Our work, in contrast, focuses specifically on learning a retrieval policy that is robust to drastic changes in sequence length, a distinct and complementary objective.

**Learned Sparse Attention and KV Cache Sharing.** Two concurrent lines of work share surface-level similarities with our approach. SeerAttention (Gao et al., 2025) uses learned block-level gating to approximate dense attention efficiently, and YOCO (Sun et al., 2024) introduces a decoder-decoder architecture with a shared global KV cache. However, both focus on efficiency within existing long-context models rather than training-free length generalization. Our work investigates a distinct goal: identifying the architectural components that enable robust extrapolation from short training contexts to extreme lengths.

# C TRAINING RECIPE

## C.1 PRE-TRAINING RECIPE

Our pre-training recipe was designed for our models of approximately 240M parameters and is summarized in Table 5. All models were pre-trained on a tokenized version of the deduplicated Pile dataset (Gao et al., 2020) using the GPT-NeoX-20B tokenizer (Black et al., 2022). The models were trained for approximately 8.05 trillion tokens using a global batch size of 1.05 million tokens. We used the AdamW optimizer (Loshchilov & Hutter, 2019) with a cosine learning rate schedule, a peak learning rate of $1 \times 10^{-3}$, and a minimum learning rate of $2 \times 10^{-4}$.

## C.2 SUPERVISED FINE-TUNING RECIPES

Following pre-training, separate fine-tuning runs were conducted for the RULER and BabiLong benchmarks. All SFT runs were initialized from their respective pre-trained checkpoints.

**SFT for RULER.** The SFT data was constructed dynamically. For each sample, one of the three RULER sub-tasks (S-N, MQ-N, or VT) was randomly selected. The task's "needle" was then inserted at a random position within a background context sampled from the Pile. The model was trained to generate the correct answer. This stage ran for approximately 1.0 trillion tokens with a peak learning rate of $2 \times 10^{-4}$.

**SFT for BabiLong.** In a similar fashion, the SFT data for BabiLong was created by dynamically embedding tasks from the original bAbI dataset (Tasks 1-20) into a background context from the Pile. For each training sample, one of the 20 bAbI tasks was randomly chosen, and its factual sentences were embedded into a context. The model was then trained on the associated question-answering pair. The hyperparameters were identical to the RULER SFT, with the primary difference being a slightly longer training schedule of approximately 1.07 trillion tokens.

| Hyperparameter | Pre-training | SFT (RULER) | SFT (BabiLong) |
|---|---|---|---|
| *Common Parameters* | | | |
| Model Parameters | | $\approx$ 240M | |
| Optimizer | | AdamW | |
| LR Schedule | | Cosine with 2% warmup | |
| Global Batch Size | | 1.05 Million tokens | |
| *Stage-Specific Parameters* | | | |
| Training Tokens | $\approx$ 8.05 Trillion | $\approx$ 1.0 Trillion | $\approx$ 1.07 Trillion |
| Peak Learning Rate | $1 \times 10^{-3}$ | $2 \times 10^{-4}$ | $2 \times 10^{-4}$ |
| Minimum Learning Rate | $2 \times 10^{-4}$ | $4 \times 10^{-5}$ | $4 \times 10^{-5}$ |

Table 5: Key hyperparameters for Pre-training and SFT stages.

# D HYBRID ARCHITECTURES WITH SSM

The recent success of hybrid architectures, which synergistically combine different components to balance performance, efficiency, and long-context reasoning, marks a significant trend in sequence modeling. This paradigm has been prominently validated in models built upon the Mamba architecture, such as Jamba (Lieber et al., 2024), Zamba/Zamba2 (Glorioso et al., 2024b;a), Samba (Ren et al., 2025), and others (Tencent Hunyuan Team, 2025; Soman et al., 2024; Wang et al., 2024). The principle extends to hybrids combining attention with linear RNNs or other mechanisms, as seen in Griffin (De et al., 2024), RWKV-X (Hou et al., 2025), RecurrentGemma (Botev et al., 2024), and large-scale MoE models (MiniMax, 2025; Jamba Team, 2024).

A notable model in this space is RAMba (Hu et al., 2025a), which integrates an SSM with a HSA and demonstrates 64x length generalization on RULER. We hypothesize that our architectural principles can further enhance the performance of such hybrid models. To test this, we apply our components

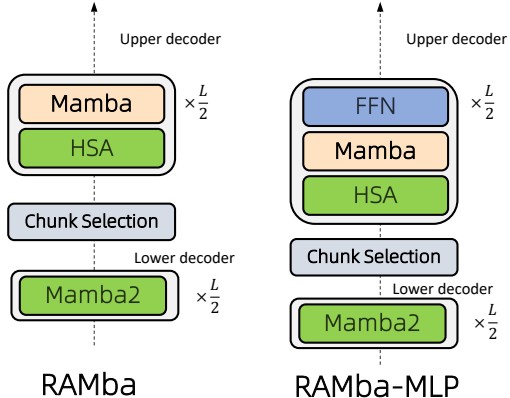

Figure 5: **RAMba Architectures.** An MLP is inserted into upper decoder layer in RAMB-MLP as Mamba does not have explicit MLP layer.

| Model | Enc.Layers | CLS | Bypass | S-N | MQ-N | VT | Avg. | S-N | MQ-N | VT | Avg. | S-N | MQ-N | VT | Avg. |
|---|---|---|---|---|---|---|---|---|---|---|---|---|---|---|---|
| | Model Specs | | | ctx-len=4K | | | | ctx-len=32K | | | | ctx-len=128K | | | |
| Mamba-2 | 0 | — | — | **99.54** | 15.96 | 80.80 | 65.43 | 1.48 | 1.49 | 0.40 | 1.12 | 0.00 | 0.00 | 0.00 | 0.00 |
| Mamba-2 w/ NSA | 0 | — | — | 64.94 | 1.48 | 35.62 | 34.01 | 4.48 | 0.00 | 5.97 | 3.48 | 0.00 | 0.00 | 15.15 | 5.05 |
| RAMba *(original)* | 2 | no | no | 90.17 | 76.16 | 94.81 | 87.05 | 82.84 | 74.63 | 88.81 | 82.09 | 84.85 | 72.73 | 96.97 | 84.85 |
| RAMba | 2 | yes | no | 94.06 | 78.20 | 94.99 | 89.08 | 87.31 | 84.33 | **98.51** | 90.05 | 81.82 | 84.85 | 90.91 | 85.86 |
| RAMba-MLP | 0 | no | yes | 84.69 | 55.47 | 71.43 | 70.53 | 85.82 | 44.03 | 35.07 | 54.97 | 75.76 | 15.15 | 15.15 | 35.35 |
| RAMba-MLP | 2 | no | yes | 95.08 | 79.04 | 94.99 | 89.70 | 94.78 | 60.45 | 85.82 | 80.35 | **100.00** | 45.45 | 81.82 | 75.76 |
| RAMba-MLP | 2 | yes | no | 93.32 | 81.35 | **96.66** | 90.44 | 84.33 | 78.36 | 96.27 | 86.32 | 84.85 | 66.67 | 96.97 | 82.83 |
| RAMba-MLP *(ours)* | 2 | yes | yes | 95.36 | **83.95** | **96.64** | **91.98** | **97.76** | **88.06** | 97.76 | **94.53** | **100.0** | **87.88** | **100.0** | **95.96** |
| Models | Enc | CLS | Bypass | S-N | MQ-N | VT | Avg. | S-N | MQ-N | VT | Avg. | | | | |
| | Model Specs | | | ctx-len=1M | | | | ctx-len=8M | | | | | | | |
| RAMba *(original)* | 2 | no | no | 33.66 | 51.49 | 43.56 | 42.90 | 6.93 | 24.75 | 22.77 | 18.15 | | | | |
| RAMba | 2 | yes | no | 57.43 | 52.48 | 32.67 | 47.53 | 17.82 | 22.77 | 17.82 | 19.47 | | | | |
| RAMba-MLP | 0 | no | yes | 68.32 | 5.94 | 30.69 | 34.98 | 48.51 | 1.98 | 16.83 | 22.44 | | | | |
| RAMba-MLP | 2 | no | yes | 92.17 | 14.81 | 60.00 | 55.66 | 80.00 | 10.00 | 45.00 | 45.00 | | | | |
| RAMba-MLP | 2 | yes | no | 48.56 | 65.49 | 48.18 | 54.08 | 15.17 | 48.28 | 10.34 | 24.60 | | | | |
| RAMba-MLP *(ours)* | 2 | yes | yes | **100.00** | **80.00** | **72.00** | **84.00** | **90.00** | **72.00** | **63.64** | **75.21** | | | | |

Table 6: **RULER results for the RAMba hybrid architecture.** The results confirm that the design principles identified for DRT generalize effectively to an SSM-based model. Our full RAMba configuration (w/ Encoder, CLS, Bypass) once again achieves the best performance, particularly at extreme context lengths ($\geq$ 1M), validating the critical and synergistic role of these components across different local architectures.

to the RAMba architecture. The original RAMba design does not include a distinct MLP block due to Mamba's block design, which is necessary for a controlled study of our Bypassing Residual Path, as shown in Figure 5. We therefore introduce an MLP after each HSA block, enabling us to isolate the effects of our proposed encoder, CLS token, and bypass mechanism on a state-of-the-art SSM-based foundation. We systematically investigate on RULER, detailed results in Table 6.

# E  EFFICIENCY ANALYSIS

In this section, we provide a detailed analysis of the computational efficiency of our proposed Hierarchical Sparse Attention (HSA) mechanism, including theoretical complexity bounds and empirical measurements of inference latency.

## E.1  COMPLEXITY ANALYSIS

Consider a model with HSA operating on a context of length $L$, with chunk size $C$, top-$K$ selection, and hidden dimension $d$. We analyze the complexity of each component:

**Chunk Encoder.** The chunk encoder processes each chunk of size $C$ independently. With $L/C$ chunks in total, the complexity is:

$$O\left(\frac{L}{C} \cdot (4Cd^2 + 2C^2d)\right) = O\left(L \cdot (4d^2 + 2Cd)\right), \tag{7}$$

which scales linearly with context length $L$.

**Top-$K$ Chunk Selection.** The top-$K$ selection computes scores between the query and each chunk's landmark representation. For a single decoding step, the complexity is:

$$O\left(\frac{L}{C} \cdot d\right). \tag{8}$$

The cumulative complexity across all $L$ steps is:

$$O\left(L \cdot \frac{L}{C} \cdot d\right) = O\left(\frac{L^2d}{C}\right). \tag{9}$$

**Hierarchical Sparse Attention.** HSA computes attention over the selected $K$ chunks, each of size $C$. For a single step:

$$O\left(K \cdot (4Cd^2 + 2C^2d)\right). \tag{10}$$

The cumulative complexity for all $L$ steps is:

$$O\left(L \cdot K \cdot (4Cd^2 + 2C^2d)\right). \tag{11}$$

**Discussion.** Only the chunk selection component involves a quadratic term with respect to $L$. However, compared to full attention's $O(L^2d)$ complexity, this term is reduced by a factor of $1/C$. Furthermore, chunk selection is performed *only once* in the entire model (at a single layer), making its quadratic contribution minimal in practice.

### E.2 INFERENCE EFFICIENCY

To empirically validate the efficiency of our proposed architectures, we measure inference latency for generating 100 tokens across varying prompt lengths. All experiments use a batch size of 16, and we report the time cost excluding the initial prefilling phase.

Table 7: **Inference Time Comparison.** Time cost (in seconds) for generating 100 tokens at different prompt lengths. RAMba (Mamba-2 + HSA) maintains near-constant latency regardless of context length, while full-attention Transformers exhibit substantial degradation.

| Model | 4K ↓ | 16K ↓ | 64K ↓ |
|---|---|---|---|
| Transformer (Full Attention) | 2.26 | 8.90 | 32.12 |
| Mamba-2 | 2.92 | 2.82 | 2.84 |
| RAMba (Mamba-2 + HSA) | 3.14 | 3.05 | 2.76 |

As shown in Table 7, both Mamba-2 and RAMba exhibit nearly constant inference time across context lengths from 4K to 64K, whereas the full-attention Transformer's latency increases by over $14\times$. Notably, RAMba incurs only a marginal overhead compared to the base Mamba-2 model while providing the additional capability of lossless long-range memory retrieval through HSA. Similarly, SWA+HSA benefits from the linear scaling properties of sliding window attention, with HSA adding only acceptable computational overhead. These results demonstrate that our proposed architectures are highly efficient for practical deployment in long-context scenarios.

## F  LLM USAGE STATEMENT

In accordance with the disclosure made during the submission process, large language models (LLM) were utilized as a writing assistant to aid in polishing the text of this manuscript. The LLM's

role was strictly limited to improving the clarity, conciseness, and narrative flow of author-written content. All scientific contributions, including the core research ideas, experimental design, and analysis of results, are the sole work of the human authors. The authors have reviewed and edited all text and take full responsibility for the final content of this paper.

