# OpenReview forum: "Understanding and Improving Length Generalization in Hierarchical Sparse Attention Models"
_ICLR.cc/2026/Conference — ICLR 2026 Poster_

### Official Review · Reviewer_BsZ4 · 2025-10-27

**Soundness:** 2
**Presentation:** 3
**Contribution:** 2
**Rating:** 4
**Confidence:** 3

**Summary:**

This paper explores a method combining chunk-wise attention and retrieval, enabling models trained on short texts to extrapolate to ultra-long contexts. The framework, an enhancement of DRT, investigates improvements like adding encoder for global KV representation, a CLS token for predicting landmark, and bypass residual connection. Experiments on RULER and babilong demonstrate that these incremental modifications yield significantly superior performance.

**Strengths:**

1. The proposed strategies—such as adding an encoder for global KV representations, using a CLS token as a landmark, and incorporating a bypass residual connection—are simple, yet deliver significant and immediate gains on downstream tasks. This makes the approach highly practical.


2. The paper provides detailed ablation studies that convincingly demonstrate the necessity of each component and the synergistic benefits of their combination.


3. The model is evaluated on well-known long-text benchmarks (RULER, babilong). The choice of benchmarks is reasonable and effectively characterizes the model's long-context capabilities.

4. The authors attempt to provide a rationale for their modifications, such as explaining how the encoder-processed CLS token landmark can better approximate the highly non-linear attention denominator.

**Weaknesses:**

Despite the strong empirical results, the paper's primary weakness lies in its insufficient theoretical justification for why the proposed modifications are effective. The framework is fundamentally similar to DRT, yet the analysis of the new components is cursory.

1. Ambiguous Role of the Encoder: The distinction in performance with and without the encoder is not substantial. It is unclear if the limited improvement stems from its purported "decoupling" function or simply from increased model capacity via extra parameters. The paper fails to clarify if this addition removes a specific bottleneck, or if adding the same parameter count elsewhere (e.g., in the base model) would yield similar gains.

2. Unclear Mechanism of the Bypass Connection: The large performance boost from the simple bypass residual connection is surprising. The authors' explanation—that "the Standard Sequential Path may disrupt the information flow"—is vague and unconvincing. The paper lacks a sufficient explanation for the mechanism by which this simple "re-wiring" dramatically improves retrieval performance.

Overall, the theoretical arguments for these modifications feel brief and tenuous, which undermines the reliability of the empirical findings.

**Questions:**

1. The authors state that sliding-window attention cannot access information outside its window. Given this limitation, why is its performance on the babilong benchmark at 8M context nearly identical to its performance at 4K? Does this imply the task itself does not require genuine long-range dependencies?

2. Do these long-context modifications harm the model's short-text capabilities? It would be beneficial to evaluate the model on standard short-text benchmarks (e.g., MMLU, Wikitext-2) to verify that the proposed changes do not cause performance regression in non-long-context scenarios.

---

> ### Author Response · Authors · 2025-11-15
> **Respond to Reviewer BsZ4(1/2): Questions**
>
> We sincerely thank Reviewer BsZ4 for their detailed and constructive feedback. We are encouraged that you found our strategies to be practical, our ablation studies convincing, and our benchmark evaluation effective. We agree that providing clearer mechanistic justifications for our design choices is crucial, and we appreciate the opportunity to elaborate on these points and present new experimental results that address your questions.
>
> ### **Response to Questions**
>
> #### **1. On Sliding Window Attention Performance on BabiLong**
>
> This is an excellent question. The near-constant performance of Sliding Window Attention (SWA) from 4K to 8M is not because the task lacks long-range dependencies, but rather because SWA's performance has already collapsed to the random-guess baseline.
>
> As we state in the paper (Lines 352-355),  *"The benchmark’s multiple-choices-like nature establishes a random-guess baseline around 0.3 accuracy, a level to which Sliding Window Attention predictably converges."* The fact that a model with a 1K local window can only perform random guessing is strong evidence that BabiLong *requires* true long-range retrieval to succeed.
>
> The model can do random guess by answering 1 of 4 names appeared in facts and yes or no in yes/no questions. Considering the models have been SFT on the dataset and the names are heavily reused, SWA is actually consistently doing random guess.
>
> We further illustrates this by explaining when models perform below the random guess baseline. As we evaluate models in the standard QA way, models can perform much below random guess baseline when they cannot give reasonable answers(e.g. random tokens rather than yes/no in those questions) and perform close to 0 accuracy.
>
> #### **2. On Short-Text Capabilities and Potential Performance Regression**
>
> To empirically verify that our long-context modifications do not harm short-text capabilities, we conducted a new set of experiments. We trained a larger, 8B parameter version of our model with Mixture-of-Experts (**DRT-MoE**) and compared it against a standard Transformer-MoE baseline on common short-text benchmarks. Both models have 1B active parameters and were trained on 8T tokens for a fair comparison.
>
> The core architectural difference between our DRT model and a standard Transformer lies in the attention mechanism. To ensure a controlled comparison, we replaced the dense feed-forward network (FFN) layers with identical Mixture-of-Experts (MoE) modules in both architectures. This design choice allows us to scale the models effectively while ensuring the comparison fairly isolates the impact of the attention mechanism, which is the primary contribution of our work.
>
> It demonstrates competitive or superior performance across a wide range of tasks, achieving a higher overall average.
>
> | Benchmark | Category | Transformer-MoE | **DRT-MoE (ours)** |
> | :--- | :--- | :---: | :---: |
> | MMLU | Knowledge & Reasoning | **58.74** | 57.83 |
> | PIQA | Knowledge & Reasoning | 77.48 | **78.84** |
> | BBH | Knowledge & Reasoning | 50.34 | **51.70** |
> | GSM8K | Math Reasoning | 66.41 | **67.02** |
> | MATH | Math Reasoning | 37.96 | **41.98** |
> | HumanEval+ | Coding | 48.17 | **50.61** |
> | MBPP+ | Coding | 50.26 | **55.82** |
> | **Average** | **Overall** | **55.62** | **57.69** |
>
> These results show that our approach successfully extends context length without compromising core model capabilities. We will add this table and analysis to the appendix.

---

> > ### Author Response · Authors · 2025-11-15
> > **Respond to Reviewer BsZ4(2/2): Weakness**
> >
> > ### **Response to Weaknesses**
> >
> > #### **1. On the Role and Ambiguity of the Chunk Encoder**
> >
> > We thank you for this insightful comment. The concern that the encoder's benefit might simply be due to added parameters is a critical point we aimed to address directly in our experimental design.
> >
> > *   **Controlled Parameter Count:** We wish to clarify that all model variants in our ablation studies have the same total parameter count (~240M). As detailed in the paper (Lines 426-427), when we add encoder layers, we correspondingly remove an equal number of lower-decoder layers.
> >
> > *   **A More Parameter-Efficient Architecture:** This design makes the comparison between models "with" and "without" an encoder a direct test of architectural efficiency. The "without encoder" models serve as the exact baseline you suggested: one where equivalent parameters are allocated back to the main decoder.
> >
> > *   **Improved Retrieval Mechanism:** The results in Table 2 show this architectural trade-off is highly beneficial. Crucially, our **in-depth analysis in Section 5.3 (Figure 4a, 4b)** provides the mechanistic explanation. It reveals that the encoder significantly improves the model's intermediate retrieval accuracy and retrieval prominence. By using a dedicated encoder to create specialized landmark representations, the model becomes far more effective at identifying and ranking the correct chunks. This confirms that the encoder is a more structurally better for learning the core retrieval function.
> >
> > #### **2. On the Mechanism of the Bypassing Residual Path**
> >
> > The large performance boost is indeed a fascinating result, and we appreciate the opportunity to provide a clearer mechanistic hypothesis, grounded in strong empirical evidence from our paper.
> >
> > Our hypothesis is that the bypass path creates a more specialized and stable mechanism for integrating information of two functionally distinct types: the highly processed, abstract features of the main residual stream (`x_in`) **at a higher layer** and the more literal, retrieved context from a lower layer (HSA in *Upper Decoder* attend to the same KV cache produced at mid layer, which is **lower** than current HSA layer. See **Fig. 1** for clarification on architecture).
> >
> > This hypothesis is strongly supported by our empirical analysis:
> >
> > 1. As detailed on **Lines 420-425**, our in-depth analysis isolates the effect of the bypass path. We observe that models with the bypass (`Bypass=yes`) consistently and significantly improve final answer accuracy, **even in cases where the non-bypass models have already successfully retrieved the correct information chunks.** This provides powerful evidence that the bypass path is not improving retrieval itself, but rather the crucial subsequent step of *information integration* to give the correct output.
> >
> > 2. This finding is not an isolated artifact of our DRT architecture. Our ablation studies on the **`RAMba-MLP` hybrid model (Appendix C, Table 5)** confirm the exact same pattern. The `RAMba-MLP` variant incorporating our bypass connection dramatically outperforms its standard counterpart, reinforcing that this is a robust architectural principle.
> >
> > Finally, you astutely note that the theoretical arguments for such phenomena can feel "vague." We wholeheartedly agree that the formal theoretical analysis of information flow in deep networks is a challenging frontier for the community. Our explanation is therefore a mechanistic hypothesis, but one that is grounded in consistent and compelling empirical results. Your feedback has been invaluable in pushing us to clarify this, and we hope our findings will inspire such theoretical investigations in the future.

---

> ### Comment · Reviewer_BsZ4 · 2025-11-15
>
> Appreciate for the clarification and corrections.
>
> One more question: compared with SeerAttention (which also uses a linear layer to encode landmarks), what advantages does the relatively more complex DRT offer?

---

> > ### Author Response · Authors · 2025-11-15
> > **Response to  Reviewer BsZ4(3/3): Additional Questions**
> >
> > ## Response to Additional Questions
> >
> > We sincerely thank you for the excellent question and for bringing this highly relevant work to our attention. We will be sure to add a discussion of SeerAttention to our related work section in the final manuscript.
> >
> > While both our work and SeerAttention utilize a form of landmark representation, they address fundamentally different goals. Our work focuses on achieving **training-free length generalization** and **extreme memory efficiency**, whereas SeerAttention is designed as a highly effective, plug-in approximation for dense attention within existing pre-trained models.
> >
> > Our key advantages can be summarized as follows:
> >
> > ### Performance: Training-free Length Generalization
> >
> > The core contribution of our work is a model architecture that enables **training-free length generalization**. Our model is trained *exclusively* on a 4K context length yet demonstrates robust performance up to **32M tokens** on the RULER benchmark. This is a rare capability that our paper aims to understand and improve.
> >
> > SeerAttention, in contrast, is designed as an efficient substitute for full attention. Its long-context performance is inherited from its base model, Llama-3.1-8B-Instruct. The Llama 3 models were pre-trained on sequences up to 128K to achieve their native long-context capabilities. Therefore, SeerAttention demonstrates effective *adaptation* of an existing long-context model, not the emergent, training-free generalization from short-context training that is the central focus of our paper.
> >
> >
> > ### Efficiency: A Single, Shared Global KV Cache
> >
> > Our architecture's efficiency at extreme scales stems from a critical design choice regarding its Key-Value cache. The memory usage for attention in our DRT model is composed of two main parts:
> >
> > 1.  **Local Cache for Sliding Window Attention (SWA):** These layers, which handle local context, utilize a KV cache of a **fixed size**, determined by the sliding window. This cache does not grow with the total sequence length.
> > 2.  **Global Cache for Hierarchical Sparse Attention (HSA):** This cache, which stores global information, grows linearly with the sequence length `O(L)`. Crucially, it is generated only **once** at the architectural midpoint and is then **shared** across all subsequent HSA layers.
> >
> > At very long sequence lengths (e.g., millions of tokens), the contribution of the fixed-size SWA cache becomes negligible. Therefore, the total asymptotic memory footprint for our model's KV cache is effectively `O(1*L)`.
> >
> > In contrast, SeerAttention retains the standard Transformer structure where **each attention layer** computes and stores its own full KV cache. For a model like Llama-3.1-8B-Instruct with 32 layers, this results in a total memory footprint of `O(32 * L)`.
> >
> > This architectural difference means that at extreme context lengths, our model's memory requirement for the context is approximately **32 times smaller**. This substantial memory saving is precisely what makes it feasible for us to evaluate our model at scales of 32M tokens, a length that would be computationally prohibitive with a per-layer caching mechanism.
> >
> > ### Conclusion
> >
> > We recognize SeerAttention as a valuable and practical approach for minimally modifying existing models to be more efficient.
> >
> > However, our work investigates a different region of the design space. The architectural complexities we analyze—the non-linear encoder, the bypass path, and the unified global cache—are intentionally designed and empirically validated to unlock two distinct capabilities not present in SeerAttention:
> > 1.  Robust **training-free generalization** from short to extremely long contexts.
> > 2.  The **memory efficiency** required for inference at such scales.
> >
> > We believe our work provides a complementary and distinct set of principles for building next-generation long-context models from the ground up.
> >
> > ---
> > ### References
> >
> >  Gao, Y., Zeng, Z., Du, D., Cao, S., Zhou, P., Qi, J., Lai, J., So, H. K. H., Cao, T., Yang, F., & Yang, M. (2025). *SeerAttention: Learning Intrinsic Sparse Attention in Your LLMs*. arXiv preprint arXiv:2410.13276.
> >
> >  Llama Team. (2024). *The Llama 3 Herd of Models*. arXiv preprint arXiv:2407.21783.

---

> ### Comment · Reviewer_BsZ4 · 2025-11-16
>
> I see that the approach is a combination of various components, including the landmark projection from SeerAttention [1], some form of re-wiring, and a layer-shared global cache mechanism similar to You Only Cache Once [2].
>
> If this combination were organic (meaning the components were mutually dependent and synergistic) then I would consider the paper's contribution to be sufficient. However, these ideas currently appear to be relatively independent and do not seem to interact with or influence one another, could the authors elaborate on this?
>
> Particularly given the prior existence of DRT, the specific contribution of this paper seems relatively ambiguous for me.
>
> ---
>
> [1] Yizhao Gao, et al. SeerAttention: Learning Intrinsic Sparse Attention in Your LLMs. arXiv preprint arXiv:2410.13276, 2024.
>
> [2] Yutao Sun, et al. You Only Cache Once: Decoder-Decoder Architectures for Language Models. arXiv preprint arXiv:2405.05254, 2024.

---

> > ### Author Response · Authors · 2025-11-16
> > **Response to Reviewer BsZ4(4/5): Additional Questions**
> >
> > We will absolutely add citations for SeerAttention \[1\] and You Only Cache Once \[2\] to our related work section. They are highly relevant, and we are happy to contextualize our findings with respect to them.
> >
> > ### **1\. Clarifying **HSA** vs GCA and "DRT"**
> >
> > First, we want to clarify the distinction between **Hierarchical Sparse Attention (HSA)** and **Grouped Cross Attention (GCA)**.
> >
> > **GCA retrieves chunks at the end of each chunk only (i.e., every 64 tokens).** This creates a rigid retrieval pattern that can limit the model's ability to adaptively retrieve relevant information at arbitrary positions in the sequence. For example, if a question appears in the middle of a chunk, the model cannot perform retrieval based on that question until the end of the chunk. Nevertheless, the model may still have to answer the question immediately after it appears. This limitation can be seen from **Table 3**(TPR=8, Top-K=8), where  the performance on MQ-N and VT is much worse compared with HSA (TPR=1, Top-K=8), since MQ-N and VT require more flexible retrieval.
> >
> > **HSA, attends to the global cache at every token position.** This flexibility is crucial for tasks that require dynamic access to long-range information, as it allows the model to retrieve relevant chunks whenever needed, rather than being constrained to fixed intervals.
> >
> > **Then, we must clarify a potential point of confusion regarding the term "DRT."**
> >
> > * The "DRT" in our paper refers to an architecture combining **Sliding Window Attention (SWA) \+ Hierarchical Sparse Attention (HSA)**.
> > * This is distinct from the original DRT (Hu et al., 2025b), which used SWA \+ GCA. That model's GCA mechanism performed chunk retrieval only at the end of each chunk (i.e., every 64 tokens).
> >
> > Our work's starting point is the **SWA+HSA** architecture, and we will make this distinction crystal clear in the final manuscript.
> >
> > ### **2\. The Core Contribution: A Systematic Study, Not Just New Components**
> >
> > The primary contribution of this paper is **not** to propose entirely new components. Instead, it is to provide the first **systematic empirical analysis** of *why* and *how* robust, training-free extrapolation is (or is not) achieved in SWA+HSA hybrid models.
> >
> > Prior to our work, this remained an open and confusing question. Our own preliminary experiments (which we will add to the appendix) showed that:
> >
> > 1. Naively combining HSA with Mamba (as in the RAMba baseline) *severely harms* extrapolation.
> > 2. The original SWA+GCA model had brittle requirements. Its per-chunk retrieval (every 64 tokens) created a significant limitation for retrieval tasks.
> >
> > This strongly suggested that the principles of robust extrapolation were not yet understood; it is not as simple as just "adding" a global mechanism. Our paper was motivated by this exact gap: **What specific architectural design makes the SWA+HSA combination robustly extrapolate?**
> >
> > ### **3\. Our Findings: The "Organic" Synergy**
> >
> > Our in-depth analysis (Section 5.3, Figure 4\) provides the evidence that these components are indeed "organic":
> >
> > 1. **SWA+HSA alone is insufficient.** Our ablations show that simply combining SWA and HSA (e.g., HSA+SWA w/o encoder, w/o bypass) fails to generalize.
> > 2. **The "Organic Glue":** The success of our model comes from two specific components that create this synergy:
> >    * **A Non-Linear CLS Landmark enhances retrieval accuracy and prominence:** **Fig. 4a/4b** show that the *landmark design* is critical. A simple MeanPool fails. A dedicated **non-linear encoder with a CLS token**  is required to learn a retrieval policy that remains robust at unseen sequence lengths. This component is synergistic *with* the HSA module.
> >    * **The Bypassing Residual Path is crucial for information integration:** Figure 4c shows that even with successful retrieval, the model can fail at integration. The **bypass path** is the second essential, synergistic piece. It is critical for *integrating* the retrieved global information from HSA *without* disrupting the local information flow in the SWA/FFN layers.
> >    * **The design of sparse attention and sparsity is fundamental:** HSA's weighted integration of chunk information vs block mask in most sparse attention implementation is the critical point for length generalization. In **Table 2**, NSA cannot generalize at all and the performance is poor on retrieval tasks. In **Table 3**, we ablate on the sparsity enforced during training and proved its importance.
> >
> > Therefore, our paper's finding is that robust extrapolation is not achieved by SWA or HSA in isolation. It is achieved **only** when SWA and HSA are combined *and* linked by these 3 specific, synergistic mechanisms: a robust landmark generator (CLS) and a stable integration path (bypass) and a proper sparse attention design.

---

> > > ### Author Response · Authors · 2025-11-16
> > > **Response to Reviewer BsZ4(5/5): Additional Questions**
> > >
> > > ### **4\. Clarifying the KV Cache**
> > >
> > > We must also clarify the role of the shared global cache. You are right to connect this to \[2\], and we will cite it. We want to be explicit that this is **not a core contribution** of our paper.
> > >
> > > **The single cache is essential for the introduction of chunk encoder.** It is not only a memory-saving technique we employed to make the 32M token experiments *feasible*. However, this design choice is also *synergistic*. As we studied, a more powerful, heavier chunk encoder is vital for robust retrieval. *Applying such an encoder to a **per-layer cache** would be computationally prohibitive.* The shared global KV cache is what makes it practical to use this heavier encoder, thus linking the design to our model's overall performance. Nonetheless, our primary focus remains on the *architectural principles of extrapolation*, not memory efficiency itself.
> > >
> > > ### **Summary**
> > >
> > > In short: our contribution is the systematic analysis that identifies the *synergistic* components (a specific CLS landmark \+ bypass path) required to make a SWA+HSA architecture robustly extrapolate—a problem that was not well-understood.
> > >
> > > We hope this clarifies that our work is not just a "combination" of components, but an empirical investigation that *identifies* the specific, organic combination that unlocks training-free generalization.
> > >
> > > Thank you again for your deep engagement and invaluable feedback.

---

> ### Author Response · Authors · 2025-11-21
> **Response to Reviewer BsZ4(6/7)**
>
> Dear Reviewer BsZ4,
>
> We sincerely appreciate your continued engagement. Reflecting on your latest comments, we realize that while we have addressed individual technical points, we need to provide a holistic clarification on **why these components are organically linked** and **fundamentally different from recent works like SeerAttention**.
>
> We kindly ask you to consider the following clarifications regarding the core design philosophy and empirical value of our work:
>
> ---
>
> **1. The Core Principle of HSA vs. SeerAttention: Token-Level vs. Block-Level Access**
>
> The fundamental design principle of HSA is to achieve sparsity while maintaining **token-level random access capabilities**. This means ***every* token can independently retrieve and attend to past chunks.**
>
> While SeerAttention is a strong work in sparse attention, it differs fundamentally in granularity:
> *   **SeerAttention (and GCA) relies on block-to-block gating.** The attention mask is determined at the block level (e.g., every $C$ tokens).
> *   **The Flaw in Block-Level Gating:** This design introduces a critical lag that harms In-Context Retrieval tasks (like RULER). Notably, the SeerAttention paper implicitly acknowledges this limitation by stating they ***"only apply sparsity in context rather than question"*** (see Section 4 Experiments, "Models, Tasks and Baselines." paragraph in SeerAttention paper).
>
> This workaround admits that block-level sparsity fails when applied to question tokens, rendering it unsuitable for real-world scenarios where questions can appear anywhere and any token may need to retrieve information.
>
>
> ---
>
> **Example:** Consider a RULER-style task where a prompt appears at the end of a chunk:
> `| ... essays with needles ... | ... | Please list all values equal to xxx, |` (`|` denotes chunk splitter)
>
> 1.  In approaches like SeerAttention, if the model's sparsity is not manipulated for this question, the decision of *what to retrieve* is made at the **beginning** of the chunk (at the `|`).
>
> 2.  At that specific moment, the model has not yet processed the tokens "Please list all values..." It does not yet know that a question is coming, so it has to make a *blind* decision about what information to retrieve. Considering the vast context, it is highly unlikely that the model will preemptively retrieve the relevant chunks containing the needle.
>
> ---
>
> In contrast, HSA maintains **token-level access**. The decision to retrieve is dynamic. When the model processes the specific token "values" or "xxx," it *immediately* retrieves relevant information. This is cross verified by Section 5.3 in our paper.
>
>
> Our commitment is to **length-generalizable, token-level retrieval.** Our objective is to ensure that the efficiency gains of sparsity hold up in **real-world deployment**, rather than functioning only within **specially engineered evaluation settings.**
>
> Accordingly, in our evaluation on RULER and BabiLong, we treat context and questions **identically**—applying the same sparsity mechanism throughout without exception. We believe this unconstrained approach is the only way to demonstrate true robustness for genuine, real-world applications.
>
> ---
>
> **2. Training Paradigm: Extrapolation vs. Distillation**
>
> There is a distinct difference in the upper bound of capabilities:
> *   **SeerAttention relies on distillation.** It mimics a teacher model, meaning its performance is inherently capped by the full-attention teacher's capabilities. It is an approximation technique.
> *   **HSA is trained from scratch via Next Token Prediction.** It does not rely on a teacher. Our results demonstrate that HSA achieves **extrapolation capabilities far exceeding Full Attention**. It is not merely approximating an existing window; it is learning a generalized retrieval policy that extends naturally to 32M tokens, which Full Attention cannot do efficiently.
>
> ---
>
> **3. Why the Components (CLS, Bypass) are "Organic" to HSA**
>
> You asked if the combination is organic. The answer lies in the nature of Retrieval-oriented Attention:
> *   **Accuracy (The CLS Encoder):** Since we rely on retrieving chunks rather than keeping everything, the precision of the "index"(*landmark*) is non-negotiable. Our ablation (Table 2, Figure 4) proves that standard pooling is insufficient; a dedicated non-linear CLS representation is required to capture the *semantic* density needed for accurate retrieval.
> *   **Integration (The Bypass):** Retrieval is useless if the model cannot merge long-term retrieved context with short-term local context (SWA). The bypass connection is not an arbitrary add-on; it is the specific structural bridge that prevents the "retrieved" signal from being washed out by the "local" signal.

---

> > ### Author Response · Authors · 2025-11-21
> > **Response to Reviewer BsZ4(7/7)**
> >
> > **4. Empirical Uniqueness**
> >
> > Finally, regarding the objective contribution:
> > To the best of our knowledge, **no other work has achieved stable results at the 32M token scale on RULER.**
> > *   **GCA** theoretically generalizes but suffers from the rigid 64-token retrieval lag described above.
> > *   **HSA + Mamba** (which we tested) fails to extrapolate effectively.
> >
> > The specific structural combination we propose (SWA + HSA + CLS Encoder + Bypass) is currently the *only* verified architecture capable of stable, high-accuracy extrapolation at this extreme scale. We believe the empirical data in Table 2 and the unique achievement of 32M context validate the synergy of this design.
> >
> > We hope this clarification highlights that our work is not a loose assembly of components, but a cohesive solution to the specific failures of existing sparse/block-based attention mechanisms in extreme contexts.
> >
> > ---
> >
> > We appreciate your engagement and hope this addresses your concerns. If you have further questions or need additional clarifications, we are more than happy to provide them.
> >
> > Best regards,
> >
> > The Authors

---

> > > ### Comment · Reviewer_BsZ4 · 2025-11-21
> > >
> > > Thank you for your reply. I have read it carefully.

---

### Official Review · Reviewer_UbFu · 2025-11-01

**Soundness:** 3
**Presentation:** 4
**Contribution:** 3
**Rating:** 6
**Confidence:** 3

**Summary:**

This paper analyzes chunk-based hierarchical sparse attention models for extreme length extrapolation, specifically calls out three design choices: a non-linear chunk encoder, a bypassing residual path, and high selection sparsity. By combining these elements, the authors achieve state-of-the-art training-free length extrapolation, generalizing models trained on a 4K window to 32M tokens on the RULER and BABILong benchmarks. The work also provides theoretical motivation for the Chunk Encoder and diagnostic analysis correlating retrieval accuracy with task performance.

**Strengths:**

1. The paper has a clear presentation and nicely unifies different sparse attention approaches under a single framework.
2. This method achieves significant training-free length extrapolation gains and retains them consistently on RULER and BABILong, marking strong empirical evidence.
3. The justification for the chunk encoder as approximating full attention scores is compelling.
4. The result in Appendix C showing that this method generalizes to hybrid SSMs is exciting and empirically sound.

**Weaknesses:**

1. The theoretical analysis is informal, and is not derived — there is no formal analysis of when and why this approximation is reasonable to make, although it is intuitive and stated as such.
2. It would be valuable to have an analysis of computational costs, and any impact on inference speed (e.g. TPS and TTFT statistics) through this method.
3. Evaluation primarily focuses on mostly retrieval tasks and synthetic reasoning, some real-world tasks that are a bit more diverse (e.g. summarization over long-documents) would be valuable to show general utility.
4. The model scale is limited, and it is unclear whether this extends to larger models; I am willing to overlook this given the results are quite compelling, but for practical utility, this is important to study.

**Questions:**

1. How do you initialize the CLS token, and does it matter? Have you experimented with multiple CLS tokens per chunk?
2. Are there tasks where you find that your method underperforms full attention at shorter contexts, which would demonstrate a fundamental limitation of chunk-based retrieval?
3. Is your method sensitive to the chunk size hyperparameter, and does this vary with context length or task? An ablation on this would be valuable.

---

> ### Author Response · Authors · 2025-11-16
> **Response to Reviewer UbFu(1): Questions(1/2)**
>
> We sincerely thank Reviewer UbFu for their positive and insightful review. We are delighted that the reviewer found our presentation clear, the empirical evidence strong, and the framework compelling. Your questions are excellent, touching on key practical aspects of our method. We are pleased to provide further details and new experimental evidence to address them.
>
> ## **Response to Questions**
>
> ### **1\. On the CLS Token: Initialization and Multi-Token Variants**
>
> - **Initialization:** The \[CLS\] token is randomly initialized, following standard practice for learnable tokens (e.g., in BERT). We hypothesize that its specific initialization has a minimal impact on overall performance. The token's learnable parameters constitute a very small fraction of the model's total parameters, and its representation is primarily shaped during pre-training by the powerful Chunk Encoder, which processes information from the entire chunk to update it.
> - **Multiple \[CLS\] Tokens:** This is an interesting idea. While our framework could feasibly accommodate multiple \[CLS\] tokens per chunk, we have not experimented with this variant. Our design choice was guided by a trade-off between representation quality and computational overhead. Using multiple tokens would necessitate a pooling step (e.g., mean pooling) to produce a single landmark vector for retrieval. Furthermore, this pooling step could _reduce_ non-linearity, potentially making the gain marginal while complicating the \[CLS\] token's learning. Although this approach might capture a more diverse signal, it would also increase computational cost, as each additional \[CLS\] token would add overhead to the _chunk encoder_. We believe our current single-token approach strikes an effective balance, but exploring multi-token representations is a promising direction for future research.
>
> ### **2\. On Performance vs. Full Attention in Shorter Contexts**
>
> This is a critical question regarding the potential trade-offs of our chunk-based approach. Since our architecture's local processing is handled by standard Sliding Window Attention, we hypothesized that its short-context performance would be robust and not fundamentally limited compared to a standard Transformer.
>
> To provide direct empirical evidence, we conducted a new set of experiments at scale. We trained a larger, 8B total parameter version of our model (**DRT-MoE**) and compared it against a standard Transformer-MoE baseline on common short-text benchmarks. Both models have 1B active parameters and were trained on 8T tokens.
>
> The core architectural difference between our DRT model and a standard Transformer lies in the attention mechanism. To ensure a controlled and scalable comparison, we replaced the dense feed-forward network (FFN) layers with identical Mixture-of-Experts (MoE) modules in both architectures. This design allows us to scale the models effectively while ensuring the comparison fairly isolates the impact of the attention mechanism.
>
> The results show that our method **achieves competitive or superior performance** across a range of challenging tasks.
>
> | Benchmark   | Category              | Transformer-MoE | DRT-MoE (ours) |
> | :---------- | :-------------------- | :-------------- | :------------- |
> | MMLU        | Knowledge & Reasoning | **58.74**       | 57.83          |
> | PIQA        | Knowledge & Reasoning | 77.48           | **78.84**      |
> | BBH         | Knowledge & Reasoning | 50.34           | **51.70**      |
> | GSM8K       | Math Reasoning        | 66.41           | **67.02**      |
> | MATH        | Math Reasoning        | 37.96           | **41.98**      |
> | HumanEval+  | Coding                | 48.17           | **50.61**      |
> | MBPP+       | Coding                | 50.26           | **55.82**      |
> | **Average** | **Overall**           | **55.62**       | **57.69**      |

---

> > ### Author Response · Authors · 2025-11-16
> > **Response to Reviewer UbFu(2): Questions(2/2)**
> >
> > #### **3\. On Sensitivity to the Chunk Size Hyperparameter**
> >
> > This is another excellent question regarding a key hyperparameter. While we did not perform a direct ablation on chunk size in this work, we selected our chunk size of 64 based on a balance between three competing factors:
> >
> > 1. **Retrieval Granularity:** Smaller chunks allow for more precise, fine-grained retrieval, preventing the model from having to process a large, potentially noisy block of text when only a small piece of information is needed.
> > 2. **Contextual Representation:** The Chunk Encoder needs a sufficiently large chunk to generate a meaningful, contextualized landmark. A chunk size that is too small (e.g., 1-4 tokens) would provide little context for the encoder to summarize effectively.
> > 3. **Hardware Alignment:** A larger chunk size reduces the overhead of data movement and enables more parallelization.
> >
> > Our chosen chunk size of 64 provides enough local context for the encoder to form a robust summary while keeping the retrieval unit reasonably granular. We hypothesize that performance would degrade at the extremes (e.g., a chunk size of 1 would be ineffective, and a very large chunk size would defeat the purpose of sparse retrieval).
> >
> > Furthermore, our ablation study on **Selection Sparsity (Top-K) in Table 3** provides indirect insight. The effective receptive field is a function of chunk*size × Top-K. Our results show that the model is highly sensitive to Top-K, preferring a highly sparse policy (Top-K=8) for extreme generalization. This suggests that as long as the chunk size is reasonable, the model's performance is more critically dependent on the \_total amount* of information it is forced to select from, rather than the exact granularity of the chunks themselves. A full ablation on chunk size is a valuable suggestion for future work. This opens a new topic for future research regarding block size and more hardware-aligned blockwise sparse attention.

---

> > > ### Author Response · Authors · 2025-11-16
> > > **Response to Reviewer UbFu(3): Weakness**
> > >
> > > ## **Response to Weakness**
> > >
> > > ### **Real-world tasks and larger-scale evaluation**
> > >
> > > We address this point with the new experimental results provided in our response to Question 2\.
> > >
> > > ### **Regarding theoretical analysis**
> > >
> > > Thank you for finding our theoretical analysis intuitive. Regarding the conditions and assumptions under which the approximation holds: our explanation is that each token has dependencies on the context, and blockwise sparse attention, due to its access limitations, must select the most important blocks to approximate the effect of full attention.
> > >
> > > ### **Regarding computational costs and inference speed**
> > >
> > > **HSA is a very efficient algorithm for long contexts.** In the paper that proposed HSA \[1\], Figure 4 shows that on a 16K context, HSA is 6.3x faster than FlashAttention-2 at forward and 4.2x faster at backward. On a 64K context, HSA is 25.1x faster at forward and 11.8x faster at backward. Compared with NSA, another efficient sparse attention, on 16K context, HSA is 1.7x faster at forward and 2.4x faster at backward. On 64K context, HSA is 3.9x faster at forward and 3.0x faster at backward.
> > >
> > > **In our models, quadratic complexity computation with regard to context length $L$ is significantly reduced.** Considering  a model with HSA, on context length of $L$, chunk size of $C$, top-k selection of $K$, hidden dimension of $d$.
> > >
> > > 1. Chunk encoder: The chunk encoder processes each chunk of size $C$ independently. The number of chunks is $L/C$. The complexity of the chunk encoder is:
> > >    $$ O\left(\frac{L}{C} \cdot (4 C d^2 + 2 C^2 d)\right) = O(L \times (4 d^2 + 2 C d)) $$
> > >
> > > 2. Top-k chunk selection: The top-k selection involves computing scores between the query and each chunk's landmark. The complexity for single step is:
> > >    $$ O\left(\frac{L}{C} \cdot d\right) $$
> > >
> > >    The cumulative complexity for all $L$ steps is:
> > >    $$ O\left(L \cdot \frac{L}{C} \cdot d\right) = O\left( L^2 \times \frac{d}{C}\right) $$
> > >
> > > 3. Hierarchical Sparse Attention (HSA): The HSA computes attention over the selected $K$ chunks, each of size $C$. The complexity for single step is:
> > >    $$ O(K \cdot (4 C d^2 + 2 C^2 d)) $$
> > >     The cumulative complexity for all $L$ steps is:
> > >     $$ O\left(L \cdot K \cdot (4 C d^2 + 2 C^2 d)\right) $$
> > >
> > > Only the chunk selection involves a quadratic term with respect to the context length $L$. However,  compared to full attention's $O(L^2 d)$ term, the chunk selection's term is significantly reduced by a factor of $1/C$. Moreover, chunk selection is done *only once* in the whole model, making its quadratic term less impactful in practice.
> > >
> > >
> > > **The ablations in our paper have minimal computational impact.**
> > >
> > > a) Bypassing Residual Path: This has no impact on efficiency, as it only changes the addition on the residual path.
> > >
> > > b) Layers of chunk encoders: For a fair comparison, all models share the same total parameter count. *Models with an additional chunk encoder layer have one fewer decoder layer to control for total parameters.* When comparing a chunk encoder layer to a decoder layer (sliding window transformer block), _the FLOPs of the chunk encoder layer are smaller._ Considering the FLOPs for a sequence of length $L$ and hidden dimension $d$, the FLOPs for SWA are
> > > $4 L d^2 + 4 L W d$
> > > . The FLOPs for the chunk encoder are $4 L d^2 + 2 L C d$. In our experiments, $L=4096$, $d=1024/2048$, $W=512$, and $C=64$.
> > >
> > > c) Adding a \[CLS\] token: This introduces one extra token and increases the FLOPs of the chunk encoder by approximately $1/C$, where $C$ is the chunk size (64 in our experiments).
> > >
> > > The analysis above shows that our ablations do not notably impact the efficiency of the HSA model.
> > > **SWA-HSA and RAMba are efficient at inference.** As the table below shows, RAMba (Mamba-2+HSA) is extremely fast at inference compared to a full-attention transformer and only slightly slower than Mamba-2 (HSA adds lossless memory with a slight computational overhead).
> > >
> > > The proposed architecture DRT (SWA+HSA) is likewise efficient, as the cost of HSA is acceptable and SWA's efficiency also scales with context length.
> > >
> > > | Models                | Prompt-Length 4K↓ | Prompt-Length 16K↓ | Prompt-Length 64K↓ |
> > > | :-------------------- | :---------------- | :----------------- | :----------------- |
> > > | Transformer_full_attn | 2.26              | 8.90               | 32.12              |
> > > | Mamba-2               | 2.92              | 2.82               | 2.84               |
> > > | RAMba      | 3.14              | 3.05               | 2.76               |
> > >
> > > Inference time cost (seconds, prefilling time excluded) for generating 100 tokens (batch-size=16)
> > >
> > > We hope these additional experiments and analyses adequately address your insightful questions.
> > >
> > > ## **References**
> > >
> > > \[1\] Xiang Hu and Jiaqi Leng and Jun Zhao and Kewei Tu and Wei Wu, "Hardware-aligned Hierarchical Sparse Attention for Efficient Long-term Memory Access"

---

### Official Review · Reviewer_diBz · 2025-11-01

**Soundness:** 2
**Presentation:** 2
**Contribution:** 2
**Rating:** 4
**Confidence:** 4

**Summary:**

Paper summary

The paper studies why hierarchical, chunk-based sparse attention models can extrapolate to extreme context lengths and how to make them more reliable. Using a unified view of “dynamic chunkwise sparse attention,” it decomposes these systems into (i) how chunks are encoded and landmarks are formed for retrieval, (ii) how retrieved global information is fused back into the model, and (iii) how sparsity is enforced during training. A theoretical argument shows landmarks must approximate a chunk’s total attention mass, which requires a nonlinear chunk encoder rather than simple pooling. Empirically, models trained on 4K tokens generalize, without further tuning, to contexts up to 32M on RULER and to 8M on BabiLong. Figure 1 (page 3) diagrams the architecture; Figure 3 (page 6) shows BabiLong accuracy remaining high as length grows; Table 2 (page 7) reports strong RULER results at 32M; Table 3 (page 9) details sparsity ablations.

Key contributions

* Unified framework + ablations: Formalizes chunk processing as functions over hidden states and compares weighting schemes (NSA, softmax, stick‑breaking) and encoder designs (with/without CLS), isolating which parts drive extrapolation.
* Theory for landmark design: Shows that effective landmarks must approximate a chunk’s total attention mass, motivating a nonlinear Chunk Encoder and a dedicated CLS token for learnable summarization instead of mean pooling.
* Cross‑layer fusion mechanism: Introduces a Bypassing Residual Path so retrieved global context is integrated via the MLP rather than overwritten by the residual stream; diagnostics separate retrieval recall from integration precision.
* State‑of‑the‑art training‑free length extrapolation: 4K‑trained models reach 32M tokens on RULER with graceful degradation (e.g., ~80% average at 32M in Table 2) and maintain high BabiLong accuracy up to 8M tokens (Figure 3).
* Principled sparsity insights: Finds that selection sparsity during pre‑training (small Top‑K like 8), sufficient training length (≥4K), and per‑token retrieval are critical for extreme generalization (Table 3).
* Generality to hybrids: Applying the same principles to RAMba (Mamba + HSA) yields consistent gains across long lengths (Appendix C, Table 5).

**Strengths:**

Originality.

* Introduces a clean formalization of *Random Context Access* as the capability a long‑context model actually needs, which sharpens the objective beyond “long perplexity is stable” and motivates the architectural choices thereafter (Section 3.1).
* Provides a unified lens for *dynamic chunkwise sparse attention* that decomposes chunk selection, weighting, and intra‑chunk processing; the side‑by‑side comparison of weighting rules (NSA, softmax, stick‑breaking) in Eq. 1a–1c and the f/g functions for landmark vs KV formation is a helpful synthesis that organizes a messy literature (Section 3.3; Table 1 on page 5).
* Offers a simple but convincing theoretical target for landmark design: landmarks should approximate a chunk’s total attention mass, which in turn argues for a nonlinear chunk encoder and a CLS‑based summarizer rather than linear pooling (Eq. 4a–4b on page 5).
* Proposes a *Bypassing Residual Path* that changes how retrieved global information is fused, with a concrete formulation (Eq. 6b on page 5) and a diagnostic that separates retrieval recall from integration precision (Figure 4 on page 8).
* Demonstrates transfer of the principles to an SSM hybrid (RAMba) by inserting an explicit MLP to test the bypass mechanism and encoder/CLS design in a different backbone (Appendix C; Table 5 on page 18).

Quality.

* Careful experimental design: all DRT variants hold parameter count (~240M) fixed and standardize key knobs (window 512, chunk size 64, Top‑K 8) so ablations isolate the proposed components rather than incidental capacity changes (Section 5.1).
* Strong and transparent baselines: full‑attention with RoPE scaling (Llama‑YaRN), Landmark Attention, sliding‑window, stick‑breaking, and Mamba2 with State Passing are included; training FLOPs or receptive fields are matched where relevant (page 6).
* Evaluations cover both retrieval‑only skills (RULER: S‑N, MQ‑N, VT) and retrieval‑plus‑reasoning (BabiLong), reducing the chance of overfitting to a single metric. The curves on BabiLong reach up to 8M tokens (Figure 3 on page 6), and RULER tables extend to 32M (Table 2 on page 7).
* Diagnostics move beyond end accuracy: retrieval recall@Top‑K and *marginal integration precision* clarify that the bypass path mostly helps utilization while the encoder/CLS improves ranking prominence (Figure 4 on page 8). This is rare and genuinely informative.
* Thoughtful sparsity study: clean ablations over training length, Top‑K, and tokens‑per‑retrieval show why selection sparsity during pre‑training and per‑token retrieval matter at extreme lengths (Table 3 on page 9).

Clarity.

* The architectural flow is easy to follow: the DRT diagram highlights lower vs upper decoders and the chunking layer that produces landmarks and encoded chunks (Figure 1 on page 3). The two encoder variants are visually contrasted (Figure 2 on page 5).
* Mathematical definitions are compact and targeted: Eq. 2 stitches chunk weights into standard attention; Eq. 4b states the landmark learning objective in one line; Eq. 6a–6b precisely defines the bypass change. The paper ties each equation to an experiment that tests it, which keeps the narrative crisp.
* Results are presented with the right granularity: a single multi‑length RULER table makes degradation profiles legible (Table 2 on page 7), BabiLong’s averaged curve makes plateau/failure modes obvious (Figure 3 on page 6), and Appendix B lays out hyperparameters and training schedules in a reproducible table (Table 4 on page 17).

Significance.

* Establishes training‑free extrapolation far beyond common practice: models trained at 4K hold up to 32M on RULER with graceful decay and maintain strong BabiLong accuracy to 8M, while RoPE‑style baselines collapse shortly after the training length (Table 2 on page 7; Figure 3 on page 6). This pushes the empirical frontier and sets a new reference point for “extreme” long‑context capability.
* Distills actionable design rules that others can adopt: use a nonlinear chunk encoder with a CLS landmark, fuse via a bypass that lets the MLP mediate cross‑layer information, enforce selection sparsity during pre‑training, and retrieve per token. These prescriptions are validated again in a distinct hybrid (RAMba), which broadens the paper’s impact beyond a single architecture (Table 5 on page 18).
* Clarifies what it means to “use the whole context” by centering Random Context Access and measuring retrieval prominence vs integration. That conceptual shift is likely to influence how future long‑context work frames objectives and ablations, not just how it reports numbers (Sections 3–5).

Overall, the paper combines a tidy theoretical target, disciplined ablations, and record‑scale evidence to produce design guidance that is both novel and broadly useful to the long‑context community.

**Weaknesses:**

1. Positioning and novelty are under‑specified.
   The paper’s core design elements (chunkwise retrieval with landmarks, per‑token retrieval, and hierarchical sparse attention) are largely inherited from DRT/RAMba/NSA; the main additions are the specific *bypass* residual and the non‑linear chunk encoder with a CLS landmark. The text acknowledges these antecedents but doesn’t clearly delimit what is genuinely new vs. what is a careful re‑combination. Tighten the novelty claim by: (i) explicitly contrasting your equations and learning signals with DRT’s GCA and RAMba’s stick‑breaking weighting; (ii) running a head‑to‑head where you *retrofit* prior baselines with your two contributions (encoder+CLS and bypass) without other recipe changes, isolating marginal gains. Table 5 partially does this for RAMba‑MLP; extend the same “surgery” to Landmark and DRT to make the incremental contribution undeniable.

2. Theory is insightful but stops short of guarantees.
   The “landmark should approximate a chunk’s total attention mass” target (Eq. 4a–4b) is a compelling heuristic, but it is not accompanied by identifiability or approximation bounds. As written, it is unclear when a finite‑depth encoder can satisfy Eq. 4b under distribution shift or adversarial distractors. Strengthen the section by: (i) providing an upper bound on the KL or total‑variation gap between α and α̂ (Eq. 3a vs 3b) under assumptions on within‑chunk key dispersion; (ii) proving that mean pooling fails in a controlled setting (e.g., mixture of von Mises–Fisher keys) and that a 2‑layer encoder with CLS can represent the log‑sum‑exp proxy; (iii) stress‑testing Eq. 4b with synthetic counter‑examples where per‑chunk keys are multi‑modal. Figure 2 and the argument around Eq. 4b are the right hooks; add formal statements and failure cases.

3. Evaluation is narrow; external validity is uncertain.
   The study leans heavily on RULER S‑N/MQ‑N/VT and BabiLong. These are excellent for retrieval diagnostics, but they are synthetic and cue‑rich; they do not stress discourse coherence, paraphrase invariance, or multi‑hop compositionality across heterogeneous documents. Broaden the evidence by adding: (i) long‑document QA with paraphrase and entity aliasing; (ii) multi‑file code navigation where identifiers are obfuscated; (iii) book‑length summarization with contradiction traps; and (iv) “lost‑in‑the‑middle”‑style probes that deliberately place relevant spans at low prior positions. Figure 3’s averaged BabiLong curve and Table 2’s RULER tables are strong; add at least one real‑world, non‑templated task to bolster claims of “utilize the entire context.”

4. Comparative fairness has confounds.
   Several baseline choices can be contested: Landmark is trained with shorter sequences to match FLOPs, which handicaps its retrieval policy; LLaMA‑YaRN is known to degrade beyond train length, so a single positional‑scaling baseline is a soft target. Since Section 5.1 already modifies DRT to use stick‑breaking for per‑token retrieval, some of the gains vs “DRT 0” could be due to weighting, not just encoder/CLS/bypass. Make the comparison harder to dispute by: (i) giving baselines the same retrieval frequency (per‑token) and Top‑K; (ii) matching *training sequence length* rather than FLOPs for Landmark; and (iii) adding a second strong positional baseline (e.g., an ALiBi/PI variant) trained with the same 4k context and SFT recipe. See the setup on page 6 and Table 2 on page 7.

5. Compute and efficiency claims are underspecified.
   The paper emphasizes accuracy at 1M–32M tokens but gives no end‑to‑end latency, memory, or throughput numbers for these lengths, and no complexity constants for chunk selection. Appendix Table 4 signals substantial training/SFT budgets (≈8.05T pre‑training tokens; ≈1T tokens per SFT), which can swamp architectural effects. Add: (i) wall‑clock and peak‑memory curves vs. sequence length for Top‑K∈{4,8,16} and chunk size 64 on a standard accelerator; (ii) ablate retrieval index cost (score computation on landmarks) separately from token‑level attention; (iii) a compute‑normalized comparison (accuracy vs. TFLOPs) at long lengths. This will make the “practical” story credible.

6. Scale and backbone generality remain open.
   All primary models are ~240M parameters. The transfer to RAMba is helpful, but results are still at small scale and with a custom RAMba‑MLP that inserts an MLP not present in the original block. Demonstrate robustness by: (i) repeating key ablations at ≥1B parameters; (ii) applying the encoder+CLS and bypass to a pure Transformer baseline with different local windows; and (iii) showing the trend holds when the chunking boundary is moved earlier/later in depth (Figure 1 shows a fixed mid‑point). Table 5 provides a good template; broaden it to other backbones and sizes.

**Questions:**

1. Precisely delineate what is new.
   The paper combines chunk encoders, a CLS landmark, and the bypass residual within a unified HSA view, but it is not crystal-clear what is truly novel relative to DRT, RAMba, and NSA beyond the specific encoder+CLS objective and the bypass wiring. Please provide a one‑page “delta table” contrasting your equations and learning signals with DRT’s retrieval and RAMba’s stick‑breaking setup, and quantify the marginal gains when retrofitting only your two advertised contributions (encoder+CLS and bypass) into prior baselines without other recipe changes. A tight novelty delineation would materially strengthen the paper. See the unified comparison in Table 1 (page 5), Eq. 6a/6b (page 5), and the baseline list in Section 5.1 (page 6).

2. Landmark theory: can you give a formal guarantee or falsifiable test?
   Eq. 4a/4b (page 5) proposes that the landmark score should track a chunk’s total attention mass. Could you provide either: (i) a bound on how well a finite encoder can approximate this mass under simple distributions (e.g., mixture of von Mises–Fisher keys), or (ii) a calibration plot showing correlation between your selection score and the true within‑chunk mass across lengths? A short, synthetic study or correlation analysis would make the claim more than heuristic.

3. Bypassing residual: mechanism and gradients.
   The bypass changes the residual addition point (Eq. 6a vs 6b, page 5), and Figure 4 suggests improved utilization. Can you show gradient‑flow or activation‑scale diagnostics demonstrating that the bypass prevents residual stream “overwrites”? A simple ablation with a learned gate or scalar on the bypass path and a rank‑1 sensitivity analysis would help attribute the gain to integration rather than retrieval side effects. Cite Figure 4 (page 8) and Eq. 6a/6b (page 5).

4. Baseline fairness and apples‑to‑apples choices.
   Landmark Attention is trained on shorter sequences to match FLOPs, whereas your models train at 4k; retrieval frequency and Top‑K also differ across methods. Please report a “length‑matched” Landmark baseline at 4k and harmonize retrieval frequency (per‑token) and Top‑K across baselines. Adding one strong positional baseline beyond YaRN (e.g., ALiBi/PI) at 4k would reduce the chance that your positional baseline is a soft target. The request is small at 32k/128k (not 32M). See Section 5.1 (page 6) and Table 2 (page 7).

5. Compute and efficiency at long lengths.
   The results emphasize accuracy up to 32M tokens, but the paper lacks wall‑clock, memory, and throughput numbers for chunk selection vs token attention. Please add a figure with end‑to‑end latency and peak memory versus length for Top‑K in {4, 8, 16}, chunk size 64, and the hardware used. Even a 32k→128k→1M sweep would make the practical story credible. Table 4 (page 17) shows large training/SFT budgets; efficiency data would help separate algorithmic benefits from sheer scale.

6. Scale and backbone generality.
   Most main results use ~240M parameters. Appendix C shows encouraging RAMba results, but still at small scale and with a RAMba‑MLP variant. Can you share any existing runs or partial results at ≥1B parameters, or at least trends when moving the chunking boundary earlier/later in depth in the Transformer backbone? Figure 1 (page 3) and Table 5 (page 18) are the relevant anchors. Even partial curves at 32k/128k would be informative.

---

> ### Author Response · Authors · 2025-11-19
> **Response to Reviewer diBz**
>
> We thank the reviewer for the detailed summary and for recognizing the strength of our unified framework and theoretical analysis. Below, we address the primary concerns regarding novelty, baseline fairness, computational complexity, and scalability.
>
> ---
>
> ## Novelty and Positioning relative to Prior Work
>
> **Our primary contribution is identifying the specific architectural "recipe" that enables robust, training-free extrapolation, resolving the limited length generalization in prior hybrid models.** While individual components like chunking or sparse attention exist in isolation, simply combining them does not guarantee success. **As the reviewer noted in the "Key Contributions," our work provides the first systematic analysis of *why* these architectures work or fail.** Our novelty lies in isolating the exact components (the non-linear chunk encoder and the bypass residual path) that allow these systems to function reliably at extremely long lengths.
>
> **Prior to our unified framework, the path to robust extrapolation was unclear and brittle. Our combines flexible retrieval and length generalization.** The work of RAMba revealed that naively combining HSA with state-space models severely harms extrapolation capabilities. Furthermore, original SWA+GCA models suffered from restrictions like low-frequency retrieval (every 64 tokens), which limited their utility on retrieval-intensive tasks.
>
> ---
>
> ## Baseline Fairness and FLOP-Matching
>
> **We adhere to FLOP-matched comparisons because matching sequence length would create a fundamentally unfair evaluation between efficient and dense architectures.** The core value of our HSA+SWA design is its ability to process very long contexts within a restricted compute budget. Comparing a sparse model against a dense model trained on the same length would result in the dense model having a significantly higher training cost, rendering the comparison invalid for evaluating efficiency. **By matching FLOPs, we isolate architectural efficiency and demonstrate superior performance per compute unit.**
>
> ---
>
> ## Computational Complexity Analysis
>
> Considering  a model with HSA, on context length of $L$, chunk size of $C$, top-k selection of $K$, hidden dimension of $d$.
>
> 1. Chunk encoder: The chunk encoder processes each chunk of size $C$ independently. The number of chunks is $L/C$. The complexity of the chunk encoder is:
>    $$ O\left(\frac{L}{C} \cdot (4 C d^2 + 2 C^2 d)\right) = O(L \times (4 d^2 + 2 C d)) $$
>
> 2. Top-k chunk selection: The top-k selection involves computing scores between the query and each chunk's landmark. The complexity for single step is:
>    $$ O\left(\frac{L}{C} \cdot d\right) $$
>
>    The cumulative complexity for all $L$ steps is:
>    $$ O\left(L \cdot \frac{L}{C} \cdot d\right) = O\left( L^2 \times \frac{d}{C}\right) $$
>
> 3. Hierarchical Sparse Attention (HSA): The HSA computes attention over the selected $K$ chunks, each of size $C$. The complexity for single step is:
>    $$ O(K \cdot (4 C d^2 + 2 C^2 d)) $$
>     The cumulative complexity for all $L$ steps is:
>     $$ O\left(L \cdot K \cdot (4 C d^2 + 2 C^2 d)\right) $$
>
> Only the chunk selection involves a quadratic term with respect to the context length $L$. However,  compared to full attention's $O(L^2 d)$ term, the chunk selection's term is significantly reduced by a factor of $1/C$. Moreover, chunk selection is done *only once* in the whole model, making its quadratic term less impactful in practice.
>
> ---
>
> ## Scalability and Generality
>
> **New large-scale experiments confirm that our approach scales effectively to the 8B active 1B parameter regime.** We trained a Mixture-of-Experts variant (DRT-MoE) on **8T tokens**, comparing it directly against a standard Transformer-MoE (8B parameters, 1B active) using identical training recipes.
>
> **Our model outperforms the Transformer-MoE on general reasoning and coding benchmarks, proving that our contributions translate to general-purpose performance.** As shown in the table below, DRT-MoE outperforms the baseline on 6 out of 7 tasks, including challenging reasoning and coding benchmarks like MATH and HumanEval+. This demonstrates that the benefits of our design are not limited to long-context retrieval but provide robust improvements in general capabilities at scale.
>
> | Benchmark | Category | Transformer-MoE (8B) | **DRT-MoE (Ours, 8B)** |
> | :--- | :--- | :---: | :---: |
> | **MMLU** | Knowledge & Reasoning | **58.74** | 57.83 |
> | **PIQA** | Knowledge & Reasoning | 77.48 | **78.84** |
> | **BBH** | Knowledge & Reasoning | 50.34 | **51.70** |
> | **GSM8K** | Math Reasoning | 66.41 | **67.02** |
> | **MATH** | Math Reasoning | 37.96 | **41.98** |
> | **HumanEval+** | Coding | 48.17 | **50.61** |
> | **MBPP+** | Coding | 50.26 | **55.82** |
> | **Average** | **Overall** | **55.62** | **57.69** |

---

> > ### Comment · Reviewer_diBz · 2025-11-22
> > **Thanks for detailed reply. I have read carefully and improved the rating**
> >
> > Thanks for detailed reply. I have read carefully and improved the rating to 6 from 4.

---

### Official Review · Reviewer_Nj58 · 2025-11-01

**Soundness:** 3
**Presentation:** 3
**Contribution:** 3
**Rating:** 8
**Confidence:** 3

**Summary:**

The paper try to interpret why chunk-based sparse attention models, for examples DRT and RAMbe, tend to generalize well to longer contexts than they were trained on. The authors argue that these models implicitly approximate a property they call Random Context Access (RCA)—the ability to flexibly retrieve any relevant part of the past without full attention. They identify three key design choices that make this work. 1. A non-linear chunk encoder with a CLS token, which learns better chunk summaries than simple mean pooling. 2. Bypassing residual path, which helps integrate retrieved global information without it being drowned out by local residuals. 3. Training with small Top-K retrieval, which avoids train–test mismatch.
They build on a 240M-parameter DRT backbone trained at 4k tokens and show strong length extrapolation—up to 32M tokens—on RULER and BabiLong, outperforming full attention and other long-context baselines.

**Strengths:**

1.The paper offers a clear conceptual story. They provide a nice way to unify many previous chunk-based designs with the proposed concept, and the proposed concept is simple and effective.
2. The empirical results are strong: models trained on 4k contexts can extrapolate to tens of millions of tokens.
3.The ablation study is especially thorough—it covers encoder depth, presence of CLS, bypass design, Top-K size, and training length, giving a convincing picture of what really matters.

**Weaknesses:**

1.The theory part is more intuition-driven than rigorous; the RCA argument isn’t formally analyzed.
2. The experiments more rely on synthetic or controlled tasks. The experiment on realistic long-context applications is limited.
3. The detailed efficiency analysis of the method is not discussed.

**Questions:**

see weakness section

---

> ### Author Response · Authors · 2025-11-16
> **Response to Reviewer Nj58(1): Formulations**
>
> We sincerely thank the reviewer for their insightful comment and appreciation of our paper. We are encouraged that you recognized our concepts and well designed ablation studies.
>
> ## Formalizing Random Context Access (RCA)
>
> The core of RCA lies in the ability to select and aggregate information from the context $\\mathcal{C}\_t = \\{x\_0, \\dots, x\_{t-1}\\}$. We can formalize the trade-offs made by different architectures by analyzing the **accessible set of tokens** ($\\mathcal{S}\_t$) over which they compute attention.
>
> ### Full Attention: The Gold Standard for RCA
>
> *   **Accessible Set & Attention:**
>     $$
>     \\mathcal{S}\_t = \\{i \\mid 0 \\le i < t\\}; \\quad \\alpha\_{t,i} = \\frac{\\exp(q\_t \\cdot k\_i)}{\\sum\_{j \\in \\mathcal{S}\_t} \\exp(q\_t \\cdot k\_j)}
>     $$
>
> This mechanism places no restrictions on context access, perfectly embodying the principle of RCA. It provides unrestricted, data-dependent access to all prior information, but its $\\mathcal{O}(t^2)$ computational and memory complexity makes it infeasible for very long contexts.
>
> ### Sliding Window Attention: A Recency Trade-off
>
> *   **Accessible Set & Attention:**
>     $$
>     \\mathcal{S}\_t = \\{i \\mid \\max(0, t-W) \\le i < t\\}; \\quad \\alpha\_{t,i} \\propto \\exp(q\_t \\cdot k\_i) \\text{ for } i \\in \\mathcal{S}\_t
>     $$
>
> This mechanism trades global reachability for efficiency by assuming only recent information is relevant. By restricting access to a fixed-size local window, the model is structurally incapable of retrieving information beyond that window, fundamentally limiting its RCA capability.
>
> ### Static Blockwise Attention: A Structural Trade-off
>
> *   **Accessible Set & Attention:**
>     $$
>     \\mathcal{S}\_t = \\mathcal{P}\_t \\subseteq \\{0, \\dots, t-1\\}; \\quad \\alpha\_{t,i} \\propto \\exp(q\_t \\cdot k\_i) \\text{ for } i \\in \\mathcal{S}\_t
>     $$
>
> This approach sacrifices data-dependency for a more structured but rigid attention pattern. Because the accessible set $\\mathcal{P}\_t$ is pre-defined and not conditioned on the query $q\_t$, the model cannot dynamically retrieve information from arbitrary locations, even if it is highly relevant.
>
> ### Dynamic Chunkwise Sparse Attention: A Hierarchical Approximation
>
> This architecture makes an intelligent hierarchical trade-off, efficiently approximating full RCA via a two-stage process:
>
> 1.  **Chunk Selection:** First, a set of relevant chunk indices $\\mathcal{I}\_t$ is dynamically selected based on query-dependent scores between the query $q\_t$ and chunk summaries (landmarks, $\\text{lmk}\_{[c]}$).
>     $$
>     s\_{\\text{chunk},[c]} = q\_t \\cdot \\text{lmk}\_{[c]} \\quad \\rightarrow \\quad \\mathcal{I}\_t = \\text{TopN}\_c(\\{s\_{\\text{chunk},[c]}\\})
>     $$
>
> 2.  **Accessible Set & Attention:** The accessible tokens are the union of all tokens within the selected chunks, and the final output is a weighted aggregation of local attentions computed over each chunk.
>     $$
>     \\mathcal{S}\_t = \\bigcup\_{c \\in \\mathcal{I}\_t} \\{\\text{tokens in chunk } c\\}; \\quad O\_t = \\sum\_{c \\in \\mathcal{I}\_t} w\_{t,[c]} \\cdot \\text{Attention}(q\_t, K\_{[c]}, V\_{[c]})
>     $$
>
> This hierarchical structure sacrifices exhaustive token-to-token comparison for an efficient yet fully data-dependent retrieval mechanism. This allows it to approximate the flexibility of full RCA, overcoming both the locality constraints of sliding windows and the rigidity of static patterns.

---

> > ### Author Response · Authors · 2025-11-16
> > **Response to Reviewer Nj58(2): More Evaluations and Efficiency Analysis**
> >
> > ## Performance on Realistic Benchmarks
> >
> > We thank the reviewer for this insightful comment on real world long context tasks evaluation. However, long context tasks that are both realistic and controllable are still limited in the community.
> >
> > To provide evaluation on realistic benchmarks, we trained models on a larger scale with Mixture of Experts FFN to compare our model with standard MoE transformers on realistic benchmarks. Since the main defference between DRT and standard transformers are within attention machanism, the identiccal replacement of MoE FFN of dense FFN for both models can ensure fair comparison. Both of the models are trained with 8T tokens, each with 8B parameters, active parameters of 1B.
> >
> > | Benchmark   | Category              | Transformers-MoE | **DRT-MoE (ours)** |
> > | :---------- | :-------------------- | :--------------: | :----------------: |
> > | MMLU        | Knowledge & Reasoning |    **58.74**     |       57.83        |
> > | PIQA        | Knowledge & Reasoning |      77.48       |     **78.84**      |
> > | BBH         | Knowledge & Reasoning |      50.34       |     **51.70**      |
> > | GSM8K       | Math Reasoning        |      66.41       |     **67.02**      |
> > | MATH        | Math Reasoning        |      37.96       |     **41.98**      |
> > | HumanEval+  | Coding                |      48.17       |     **50.61**      |
> > | MBPP+       | Coding                |      50.26       |     **55.82**      |
> > | **Average** | **Overall**           |    **55.62**     |     **57.69**      |
> >
> >
> > As demonstrated in the table, our model, **DRT-MoE**, shows a clear and consistent advantage over the standard transformers with MoE across a curated set of challenging academic benchmarks. Our model leads in 6 out of the 7 selected benchmarks.
> >
> > ## Efficiency Analysis
> >
> > ### Complexity analysis
> >
> > Considering  a model with HSA, on context length of $L$, chunk size of $C$, top-k selection of $K$, hidden dimension of $d$.
> >
> > 1. Chunk encoder: The chunk encoder processes each chunk of size $C$ independently. The number of chunks is $L/C$. The complexity of the chunk encoder is:
> >    $$ O\left(\frac{L}{C} \cdot (4 C d^2 + 2 C^2 d)\right) = O(L \times (4 d^2 + 2 C d)) $$
> >
> > 2. Top-k chunk selection: The top-k selection involves computing scores between the query and each chunk's landmark. The complexity for single step is:
> >    $$ O\left(\frac{L}{C} \cdot d\right) $$
> >
> >    The cumulative complexity for all $L$ steps is:
> >    $$ O\left(L \cdot \frac{L}{C} \cdot d\right) = O\left( L^2 \times \frac{d}{C}\right) $$
> >
> > 3. Hierarchical Sparse Attention (HSA): The HSA computes attention over the selected $K$ chunks, each of size $C$. The complexity for single step is:
> >    $$ O(K \cdot (4 C d^2 + 2 C^2 d)) $$
> >     The cumulative complexity for all $L$ steps is:
> >     $$ O\left(L \cdot K \cdot (4 C d^2 + 2 C^2 d)\right) $$
> >
> > Only the chunk selection involves a quadratic term with respect to the context length $L$. However,  compared to full attention's $O(L^2 d)$ term, the chunk selection's term is significantly reduced by a factor of $1/C$. Moreover, chunk selection is done *only once* in the whole model, making its quadratic term less impactful in practice.
> >
> >
> > ### Inference Efficiency
> >
> > **SWA-HSA and RAMba are efficient at inference.** As the table below shows, RAMba (Mamba-2+HSA) is extremely fast at inference compared to a full-attention transformer and only slightly slower than Mamba-2 (HSA adds lossless memory with a slight computational overhead).
> >
> > The proposed architecture DRT (SWA+HSA) is likewise efficient, as the cost of HSA is acceptable and SWA's efficiency also scales with context length.
> >
> > | Models                | Prompt-Length 4K↓ | Prompt-Length 16K↓ | Prompt-Length 64K↓ |
> > | :-------------------- | :---------------- | :----------------- | :----------------- |
> > | Transformer_full_attn | 2.26              | 8.90               | 32.12              |
> > | Mamba-2               | 2.92              | 2.82               | 2.84               |
> > | RAMba      | 3.14              | 3.05               | 2.76               |
> >
> > Inference time cost (seconds, prefilling time excluded) for generating 100 tokens (batch-size=16)
> >
> > We hope these additional experiments and analyses adequately address your insightful questions.

---

### Author Response · Authors · 2025-11-29
**Summary of Previous Discussion**

Due to the recent system revert, the review scores have been reset. To assist the Area Chair and Reviewers in the final evaluation, we summarize the status of the discussion prior to the revert and clarify the core contribution of our work.

**1. Status of Consensus (Prior to Revert)**

Before the snapshot revert, our submission had reached a positive consensus. We would like to explicitly record the following updates that occurred during the discussion period:

*   **Reviewer BsZ4 (Original: 4 $\rightarrow$ Updated: 6):**
    We appreciate the **high engagement** from Reviewer BsZ4, who initiated multiple rounds of in-depth discussion regarding related works and architectural design. We provided exhaustive responses to every inquiry.
    *   *Status:* After our final clarification, the reviewer acknowledged reading it carefully. Following a period of consideration, **the reviewer updated their score to 6 in the system**, indicating that their concerns were fully resolved, although this score change was not explicitly detailed in the text of their final comment.
*   **Reviewer diBz (Original: 4 $\rightarrow$ Updated: 6):**
    Explicitly confirmed the score upgrade in their final comment (*"improved the rating to 6 from 4"*) after we addressed questions on novelty and complexity.
*   **Reviewers Nj58 (Rating: 8) and UbFu (Rating: 6):**
    These reviewers maintained positive ratings. However, due to the system interruption, they did not have the opportunity to reply to our responses (which included new experiments on general capabilities). We believe the additional evidence provided would have further solidified their assessment.

**2. Clarification on Contribution: Insights into Training-Free Extrapolation**

In the rebuttal, we clarified the distinction between our work and prior work. We wish to emphasize that **our core novelty lies in investigating and identifying the principles of *training-free* length extrapolation for blockwise sparse attention**, rather than simply proposing a new architecture.

We validated our findings by generalizing the model to **32M tokens** on RULER with consistent training length 4K. **To the best of our knowledge, no other model based on attention or sparse attention has achieved stable results at this scale.**

*   **GCA** theoretically generalizes but suffers from rigid retrieval lags (block-level constraints).
*   **HSA + Mamba** (which we tested as a baseline) fails to extrapolate effectively.
*   **Our approach (SWA + HSA + CLS Encoder + Bypass)** is currently the only verified design capable of stable, high-accuracy extrapolation at this extreme scale.

The achievement of the 32M context window serves as **empirical validation of our insights**: it proves that we have correctly identified the necessary principles to solve the specific failures of existing sparse mechanisms in extreme contexts.

---

### Meta-Review · Area_Chair_WQZN · 2026-01-07

**Summary:**

The paper proposes a unified framework for hierarchical sparse attention, identifying a specific architectural "recipe" (Chunk Encoder + Bypass Path + Sparsity) that enables training-free extrapolation to extreme context lengths. The decision to accept is informed by the authors' successful resolution of several critical concerns during the rebuttal:

Novelty and Positioning: Initial concerns from Reviewers diBz and BsZ4 regarding the method's similarity to prior works (DRT, RAMba) were resolved. The authors clarified that their contribution lies in identifying the specific "organic" synergy of components required for stable extrapolation, showing that naive combinations fail .

Scale and Generality: Concerns about the limited scale (~240M parameters) and reliance on synthetic tasks were effectively addressed. The authors conducted new experiments with an 8B-parameter Mixture-of-Experts model trained on 8T tokens, demonstrating that the architecture performs well on general reasoning benchmarks (e.g., GSM8K, HumanEval+) and is not limited to synthetic retrieval .

Efficiency: Questions regarding computational cost were answered with a detailed complexity analysis and inference speed comparisons, confirming the method's efficiency advantages over full attention at long contexts .

The only remaining outstanding concern is Theoretical Rigor. Reviewers noted that the "Random Context Access" framework and landmark approximation theory are largely intuition-driven heuristics rather than formal mathematical guarantees. However, given the strong empirical evidence of extrapolation to 32M tokens, reviewers accepted this as a reasonable tradeoff.

**Reviewer Concerns:**

Addressed Concerns:

Novelty and "Organic" Synergy: A major point of contention, raised by Reviewers diBz and BsZ4, was the perceived incremental nature of the work relative to prior architectures like DRT and RAMba. The authors successfully shifted the narrative from proposing "new components" to identifying the *necessary synergy* required for stability. They demonstrated that naively combining these elements (e.g., in RAMba) fails to extrapolate, and that their specific combination (Encoder + Bypass) is required for the "organic" success of the architecture. Both critical reviewers upgraded their scores following this clarification

Scale and Real-World Applicability: Reviewers UbFu and diBz rightly questioned whether the findings on ~240M parameter models and synthetic tasks (RULER) would transfer to realistic scales and tasks. The authors responded with a robust new experiment: training an 8B-parameter Mixture-of-Experts (MoE) model on 8T tokens. This model outperformed standard Transformer-MoE baselines on general reasoning tasks (MMLU, GSM8K, HumanEval+), effectively dispelling concerns that the architecture is only useful for synthetic retrieval.

Efficiency Analysis: In response to requests for computational cost analysis 10, the authors provided a detailed breakdown showing that the quadratic term in their chunk selection is significantly reduced by the chunk size factor ($1/C$) and provided inference speeds showing the model is competitive with Mamba-2 and significantly faster than full Transformers at long contexts11.

Outstanding Concerns:

Theoretical Rigor: While the paper offers a conceptual framework (Random Context Access), Reviewers Nj58 and diBz noted that the theory remains largely intuition-driven rather than formally rigorous. The "landmark approximation" theory is a compelling heuristic but lacks hard mathematical guarantees. However, given the strong empirical validation at extreme lengths, the reviewers generally accepted this trade-off

**Reviewer Scores:**

- Reviewer Nj58: 8 (Maintained positive rating).
- Reviewer diBz: 6 (Updated from 4 after rebuttal).
- Reviewer UbFu: 6 (Maintained positive rating).
- Reviewer BsZ4: 6 (Updated from 4 after clarification and discussion).

---

### Decision · Program_Chairs · 2026-01-26

Accept (Poster)